# Concept-Driven Continual Learning

**Sin-Han Yang**                                                     *b08202029@ntu.edu.tw*
*National Taiwan University*

**Tuomas Oikarinen**                                                 *toikarinen@ucsd.edu*
*UC San Diego*

**Tsui-Wei Weng**                                                    *lweng@ucsd.edu*
*UC San Diego*

**Reviewed on OpenReview:** *https://openreview.net/forum?id=HSW49uvCNW*

## Abstract

This paper introduces two novel solutions to the challenge of catastrophic forgetting in continual learning: **Interpretability Guided Continual Learning** (IG-CL) and **Intrinsically Interpretable Neural Network** (IN2). These frameworks bring interpretability into continual learning, systematically managing human-understandable concepts within neural network models to enhance knowledge retention from previous tasks. Our methods are designed to enhance interpretability, providing transparency and control over the continual training process. While our primary focus is to provide a new framework to design continual learning algorithms based on interpretability instead of improving performance, we observe that our methods often surpass existing ones: IG-CL employs interpretability tools to guide neural networks, showing an improvement of up to 1.4% in average incremental accuracy over existing methods; IN2, inspired by the Concept Bottleneck Model, adeptly adjusts concept units for both new and existing tasks, reducing average incremental forgetting by up to 9.1%. Both our frameworks demonstrate superior performance compared to exemplar-free methods, are competitive with exemplar-based methods, and can further improve their performance by up to 18% when combined with exemplar-based strategies. Additionally, IG-CL and IN2 are memory-efficient as they do not require extra memory space for storing data from previous tasks. These advancements mark a promising new direction in continual learning through enhanced interpretability[1].

## 1 Introduction

Continual learning is a crucial aspect of machine learning, empowering models to adapt and improve as they encounter new data over time. This dynamic learning process, however, faces a significant hurdle known as "catastrophic forgetting." This phenomenon, where a model forgets previously learned information upon acquiring new knowledge, is largely attributed to the shift in input distribution with changing tasks. According to van de Ven et al. (2022); De Lange et al. (2021), there are three primary settings in continual learning: class incremental, task incremental, and domain incremental. Our focus in this paper is on the class incremental setting, identified as the most challenging setting among three due to its pronounced susceptibility to catastrophic forgetting, as detailed in van de Ven et al. (2022); Chaudhry et al. (2018); De Lange et al. (2021).

Current strategies in continual learning fall into the following main categories: **Exemplar-free** and **Exemplar-based** methods. **Exemplar-free** methods do not rely on data from old tasks, like *regularization-based* methods that introduce constraints to preserve old knowledge; *architecture-based* methods that adapt the model's structure for new tasks. On the other hand, **Exemplar-based** methods, also known as *replay-based*

---

[1]Our code is available at https://github.com/Trustworthy-ML-Lab/concept-driven-continual-learning

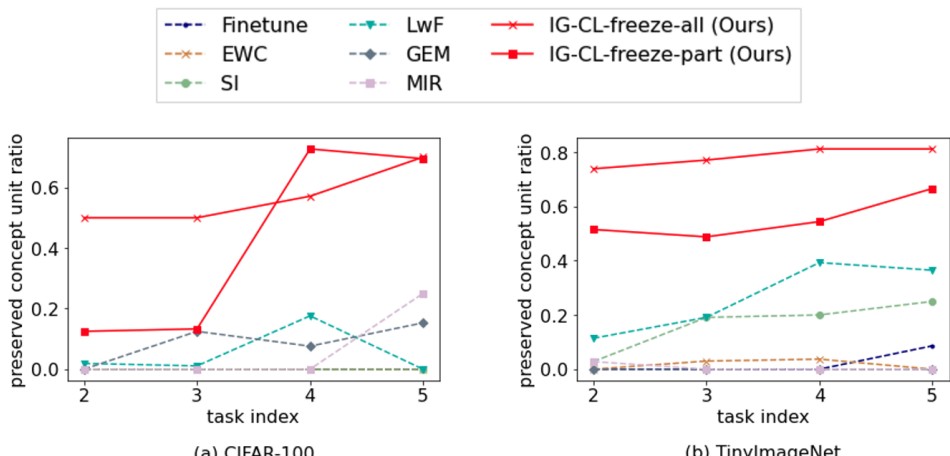

Figure 1: IG-CL's concept evolution in (a) CIFAR-100 and (b) TinyImageNet. Tasks' classes are in Appendix D.3. The results show that existing work perform poorly on preserving the learned concepts hence suffers from catastrophic forgetting problem, while the proposed IG-CL can preserve the concepts from previous classes. The values are in Table 5 of Appendix A.1.

methods, involve revisiting previous data. *Replay-based* methods generally outperform others but require extra memory for data storage. Despite existing methods show some promise, they still lack in performance and many of them still have serious catastrophic forgetting problems. This gap highlights the need for more systematic and understandable approaches in continual learning.

In parallel, there is a growing body of research dedicated to deciphering the role of neurons in neural networks. Pioneering studies, such as (Bau et al., 2017; 2020; Oikarinen & Weng, 2023; Hernandez et al., 2022; Bai et al., 2024; Oikarinen & Weng, 2024), have highlighted "concept units" – neurons that closely align with concepts easily understood by humans. Our research reveals a critical gap in current continual learning algorithms: While previous continual learning algorithms generally fail to effectively preserve these concept units, our research reveals the advantage of leveraging interpretability for mitigating catastrophic forgetting. This key finding, detailed in Figure 1, has shaped our approach to developing new continual learning frameworks.

In this paper, we present two novel frameworks, **Interpretability Guided Continual Learning** (IG-CL) and **Intrinsically Interpretable Neural Network** (IN2), to address the issue of catastrophic forgetting in continual learning. Unlike existing approaches that lack interpretability, our methods stand out by incorporating interpretability, enabling systematic management of human-understandable concepts within the model, which enhances knowledge retention from prior tasks. Our contributions are the following:

- Our first method, IG-CL, uses interpretability tools (Bau et al., 2020; Oikarinen & Weng, 2023) to guide neural networks in continual learning. IG-CL has proven to be effective in our extensive experiments, outperforming existing methods by achieving up to 1.4% higher average incremental accuracy.

- Our second method, IN2, is a novel framework based on the Concept Bottleneck Model (CBM) (Koh et al., 2020; Oikarinen et al., 2023), tailored for continual learning. IN2 is engineered to maintain and adapt concept units corresponding to both old and new tasks, further reducing average incremental forgetting by up to 9.1% compared to existing methods. Different from IG-CL, IN2 modifies the model's architecture and inherit interpretability from CBM.

- To our best knowledge, IG-CL and IN2 are the first general approach to make the learned knowledge and the training process transparent and interpretable in the continual learning setting.

Our primary aim is to leverage interpretability to provide transparency and control over retained knowledge and the training process, addressing gaps in the current literature. IG-CL and IN2 bridge interpretability and

Table 1: Comparison of our method against existing methods for continual learning.

| | (I) Scalability | | (II) Flexibility | (III) Interpretability |
|---|---|---|---|---|
| Method: | Only need one backbone | Without memory buffers | For any architecture | |
| **Exemplar-free** | | | | |
| EWC (Kirkpatrick et al., 2017) | **Yes** | **Yes** | **Yes** | No |
| SI (Zenke et al., 2017) | **Yes** | **Yes** | **Yes** | No |
| LwF (Li & Hoiem, 2017) | **Yes** | **Yes** | **Yes** | No |
| Adam-NSCL (Wang et al., 2021) | **Yes** | **Yes** | **Yes** | No |
| SSRE (Zhu et al., 2022) | **Yes** | **Yes** | **Yes** | No |
| DEN (Yoon et al., 2018) | **Yes** | **Yes** | **Yes** | No |
| ICICLE (Rymarczyk et al., 2023) | **Yes** | **Yes** | No | △ (part-based prototype) |
| **Exemplar-based** | | | | |
| GEM (Lopez-Paz & Ranzato, 2017) | **Yes** | No | **Yes** | No |
| MIR (Aljundi et al., 2019) | **Yes** | No | **Yes** | No |
| DER (Yan et al., 2021) | No | No | No | No |
| IG-CL (**Ours**) | **Yes** | **Yes** | **Yes** | **Yes, text-based concept (more general)** |
| IN2 (**Ours**) | **Yes** | **Yes** | **Yes** | **Yes, text-based concept (more general)** |

continual learning from two perspectives. IG-CL brings interpretability to continual learning by using external interpretability tools to guide models. On the other hand, IN2 tailor interpretable models like CBM for continual learning by leveraging their own interpretability. While our focus is on proposing a new framework to design continual learning algorithms based on interpretability, we observe that our methods often surpass existing ones. For instance, our approach outperforms exemplar-free methods in average incremental accuracy and is competitive with exemplar-based methods. When combined with exemplar-based strategies, our methods enhance their performance by up to 18%. Additionally, our methods are memory-efficient, requiring no extra memory space for storing data from previous tasks. The benefits of our methods are summarized in Table 1. Interpretability is key distinguishing feature of our approach compared to existing methods.

## 2 Background and Related Work

### 2.1 Continual Learning

To mitigate catastrophic forgetting in continual learning, several methods have been proposed. They can be categorized into **Exemplar-free** and **Exemplar-based** methods. For **Exemplar-free** methods, it consists of two streams: regularization-based methods and architecture-based method. The key idea of regularization-based methods is to add additional terms in the loss function to constrain model parameters to not change too much from previous tasks (Kirkpatrick et al., 2017; Zenke et al., 2017; Li & Hoiem, 2017; Jung et al., 2016; Dhar et al., 2019; Castro et al., 2018; Hu et al., 2019; Lee et al., 2019; Aljundi et al., 2018; Chaudhry et al., 2018; Lee et al., 2017; Schwarz et al., 2018). On the other hand, architecture-based methods modify the model's architecture or parameters when learning new tasks, by dynamic expansion or pruning (Rusu et al., 2016; Yoon et al., 2018; Xu & Zhu, 2018; Yan et al., 2021; Li et al., 2019; Serra et al., 2018; Wang et al., 2021; Zhu et al., 2022). For **Exemplar-based** method, it is also known as replay-based methods, whose spirit is to store previous tasks' information and train the model with new tasks jointly (Lopez-Paz & Ranzato, 2017; Rebuffi et al., 2017; Chaudhry et al., 2019; Rolnick et al., 2019; Hou et al., 2019; Wu et al., 2019; Buzzega et al., 2020; Wang et al., 2022; Guo et al., 2022; Liu et al., 2021; Aljundi et al., 2019). More details of these methods is in Appendix D.2.

Meanwhile, some works focus on the theoretical aspect of continual learning (Peng et al., 2023; Peng & Risteski, 2022; Cao et al., 2022; Ruvolo & Eaton, 2013; Pentina & Urner, 2016; Chen et al., 2022; Kim et al., 2022). However, none of these methods are able to control human-interpretable concepts directly, which makes them lack interpretability. Indeed, interpretability is one of the main differences between our methods and existing methods. There are some recent works (Marconato et al., 2023; Rymarczyk et al., 2023) connects interpretability with continual learning. They focus on part-based prototype concepts or neuro-symbolic concepts. Our work focuses on text-based concepts instead, which allows more general interpretability. Meanwhile, they are only suitable for particular model architectures whereas our methods are compatible for general DNN.

### 2.2 Neuron-Level Interpretation and Concept Bottleneck Models

Several works (Bau et al., 2017; 2020; Oikarinen & Weng, 2023; Hernandez et al., 2022; Mu & Andreas, 2020; Bai et al., 2024; Oikarinen & Weng, 2024) provide automated descriptions of the roles of individual neurons in deep vision models, and do extensive studies for their methods' interpretability. Typically these methods generate a description by analyzing what kinds of inputs result in high activations for the given neuron. For example, Network Dissection (Bau et al., 2020) identifies the concepts of individual neurons by comparing the neuron's activation map to concept annotated data. A more recent work CLIP-Dissect (Oikarinen & Weng, 2023) eliminates the need of concept annotated data by leveraging the Contrastive Language-Image Pre-training (CLIP) model (Radford et al., 2021) and designing several similarity functions. Detailed introduction of CLIP-Dissect is in Appendix D.2.2. In this paper, we explore a departure from the typical emphasis on refining interpretability tools, instead utilizing them to guide continual learning processes. Our work focuses on the computer vision domain since the interpretability tools in the domain have made significant progress. It can be applied to other domains like NLP with their interpretability tools like Sajjad et al. (2022); Lee et al. (2023) potentially in the future.

Concept Bottleneck Model (CBM) (Koh et al., 2020) has a layer called Concept Bottleneck Layer (CBL) where each neuron corresponds to a human interpretable concept. The idea of CBL is to create an meaningful concept mapping from black-box representations to a layer that contains a human-interpretable concept set $C$, allowing effective intervention and explaining the model predictions. Recent works (Oikarinen et al., 2023; Yuksekgonul et al., 2023; Srivastava et al., 2024) try to address it since concept annotations is expensive and hard to collect. Specifically, Label-Free CBM (LF-CBM) (Oikarinen et al., 2023) generates concepts from task information. Detailed introduction of LF-CBM is in Appendix D.2.2. The concept set $C$ generation depends on the CBM. For LF-CBM (Oikarinen & Weng, 2023), concept sets are generated by asking large language models like GPT-3 (Brown et al., 2020) about the concepts related to the classes. For original CBM (Koh et al., 2020), concept sets has to be predefined and provided in the dataset. The primary focus of these CBM works is only on single task and does not consider the challenging continual learning setting. In contrast, our IN2 is tailored for continual learning with a new learning procedure that enables it to learn a series of tasks, which generalizes CBM-based methods to continual learning.

## 3 Our first method: IG-CL

**Overview.** To address catastrophic forgetting, we aim to manage concepts learned by models in an interpretable manner. We introduce two frameworks: **Interpretability Guided Continual Learning** (IG-CL) in this section and **Intrinsically Interpretable Neural Network** (IN2) in the next section. IG-CL uses an interpretability tool to steer the learning process, identifying and preserving previously learned concepts to prevent forgetting. This approach enhances model comprehension and has shown to improve incremental accuracy by up to 1.4%. Building on this, IN2 incorporates interpretable neurons that are systematically integrated and controlled, further reducing forgetting by up to 9.1% in our experiments. Figure 2 illustrates the key steps in IG-CL, and the details of each step are described below. IG-CL's full algorithm is summarized in Algorithm 1 in Appendix C.2.

### 3.1 Step 1: Decipher network

Initially, we use a neuron-interpretability tool to decipher the interpretable neurons in the model. Interpretable neurons are defined as those whose activations correspond closely to human-understandable concepts (e.g. "red", "stripes", "windy", etc). A model, denoted as $\theta$, consists of a feature extractor $\{W_l\}_{l=1}^{L}$, also known as a backbone, and a prediction layer $W_F$. Our goal is to preserve the important concepts learned from the previous tasks. Previous work (Bau et al., 2020) analyzes the concept neurons in each layer, and finds out that high-level concept neurons emerge in the deep layers. Low-level concepts are basic descriptions like colors, while high-level concepts are objects or components of the images. The high-level concepts are specific to the classes in the previous tasks, which needed to be preserved. Therefore, we apply the interpretability tool on the last layer of the feature extractor $W_L$ to decipher the neurons. The discovered interpretable

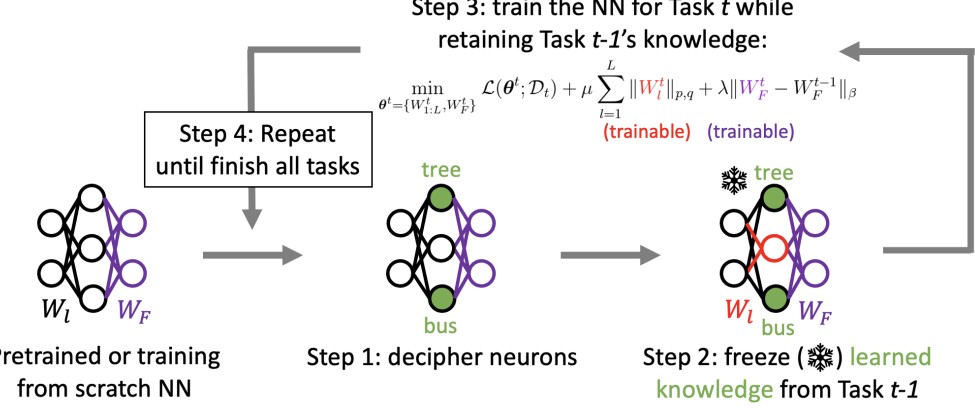

Figure 2: IG-CL's procedure.

neurons are called **concept units**. Existing neuron interpretability tools (Bau et al., 2017; Oikarinen & Weng, 2023) are directly applicable to detect the concepts of neurons in our framework.

### 3.2 Step 2: Freezing the learned knowledge

Second, we want to freeze the concept units in the model to preserve the learned knowledge. These concept units emerged in the network during learning the current task $t - 1$. To prevent catastrophic forgetting, it is desired to preserve these concept units when learning a new task $t$. To achieve this, we propose to find and freeze these task-related neurons when learning the new task $t$ by finding a sub-network which links all related neurons. In practice, this can be implemented by using Breadth First Search (BFS) from the concept unit in $W_L$ to the input layer $W_1$. Define two neurons are connected when the weight exceeds threshold $\tau$. We propose two ways to freeze the subnetworks to preserve the learned knowledge by different degrees:

- **freeze-all**  Freeze all weights connected to neurons in any concept units' subnetwork.

- **freeze-part**  Only freeze input weights to all neurons in the subnetworks, leaving output weights trainable.

We describe Step 2 mathematically in Line 7 - Line 19 of Algorithm 1 in Appendix C.2.

The **freeze-part** method leaves the output weights of the neurons in the subnetwork trainable, which may have a better ability to learn new tasks. For **freeze-all** method, it isolates the subnetworks from the rest of the model which is expected to promote less forgetting. An illustration of these two methods is in Figure 3. The experiment results in Section 5 show that the **freeze-all** method forgets less, while the **freeze-part** method has better learnability for a longer series of tasks as expected. This step is summarized in Algorithm 1 Line 8-19.

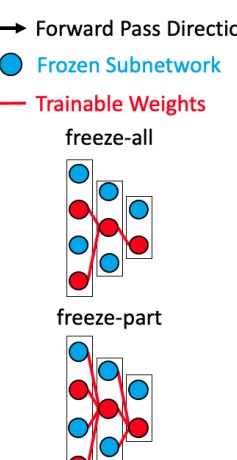

Figure 3: The illustration of two freezing subnetwork methods. Red nodes and lines stand for trainable unit and weights.

### 3.3 Step 3: Learning without forgetting

Next, we train the model on a new task $t$ with Eq. (1). Let $\mathcal{D}_t$ be the data for task $t$. We have two goals when learning task $t$: regularize (i) the size of subnetworks and (ii) $W_F$. To regularize the size of subnetworks described in Step 2, we choose $p = 2, q = 1$ and $\mu$ to regularize the number of activated units. Second,

inspired by the regularization-based methods, we train the model with regularization $\beta = 2$ on the $W_F$, which aims to prevent $W_F$ from changing too much. But different from previous regularization-based methods, the preserved knowledge in our method is interpretable (as they are detected as interpretable concept units in Step 1, which can be easily understood through the text description e.g. "*tree*", "*bus*" in Fig 2).

$$\min_{\boldsymbol{\theta}^t} \mathcal{L}(\boldsymbol{\theta}^t; \mathcal{D}_t) + \mu \sum_{l=1}^{L} \|W_l^t\|_{p,q} + \lambda \|W_F^t - W_F^{t-1}\|_{\beta} \tag{1}$$

Then, we will repeat Step 1 to 3 for task 2, 3 . . . until the last task $T$, as illustrated in Figure 2.

### 3.4 Further Discussion

**Interpretability.** Unlike previous methods in continual learning, which lack interpretability due to their inability to clearly identify retained knowledge in a human-understandable manner, IG-CL simplifies the knowledge preservation process, ensuring clarity and straightforwardness. Notably, IG-CL is designed to be compatible with existing neuron interpretability tools (Bau et al., 2017; Oikarinen & Weng, 2023), allowing it to integrate seamlessly with multiple frameworks and enhance the transparency of the learning process. Nonetheless, the selection of interpretability tools will affect IG-CL. The generalizability of interpretability tools are the key to guide and improve continual learning. Therefore, it's crucial that the interpretability tools we selected to be flexible (e.g. open-vocabulary concepts). Meanwhile, they have to be efficient so the continual learning process will not be too time-consuming. Based on these criteria, in this work we use CLIP-Dissect (Oikarinen & Weng, 2023) as the tool to detect neuron concepts.

**Ability to Share Concepts Among Tasks.** IG-CL is designed to keep and reuse the learned concepts from old tasks. Experiment in Section 5.3, Appendix A.1 and B.6 study its ability and show that it outperforms baselines by up to 9% in forward transfer metric.

**Efficiency.** IG-CL framework is light since it does not need extra memory storage to store previous tasks' samples. Previous tasks' knowledge is stored within the model. Experiment on GPU usage in Appendix C.1 shows the efficiency of IG-CL.

## 4 Our second method: IN2

In the previous Section 3, we introduced IG-CL as a novel interpretable method to identify and manage concept units in models for reducing catastrophic forgetting, marking it as a pioneering approach in the regularization-based category. Building on this, we now present the **Intrinsically Interpretable Neural Network** (IN2), a new architecture-based method that transforms any neural network into an interpretable framework designed for continual learning. IN2 uniquely maintains previously learned concept units while integrating new ones for upcoming tasks, demonstrating a reduction in average incremental forgetting by up to 9.1% over existing methods according to our experiments. Compared with IG-CL, IN2 does not need external interpretability tools since CBM provides interpretability. Figure 4 illustrates the procedure of IN2.

### 4.1 Step 0: Set up CBM

For the first task, the learning procedure is the same as training a Concept Bottleneck Model (CBM) (Koh et al., 2020) on the first single task. In the continual learning setting, after learning the task $t-1$, the CBM has a concept set $\mathcal{C}^{t-1}$ and the concept mapping $W_c^{t-1}$ in between the backbone and the prediction layer. We consider the setting that backbone $f(x)$ is frozen for reduced training cost, but our method also allows backbone being trained from scratch or finetuned. For task $t$, we create a task-relevant concept set $c_k^t$ from its class labels $k$. Creation of the task-relevant concept sets depends on the Concept Bottleneck Model (CBM). For Label-free CBM (Oikarinen et al., 2023), concept sets are generated by asking GPT-3 about the concepts related to the classes. For CBM (Koh et al., 2020), concept sets are predefined and provided in the dataset. In the following paragraphs of IN2's learning procedure, we consider the case that the model has learned $t-1$ tasks, and it is going to learn a new task $t$.

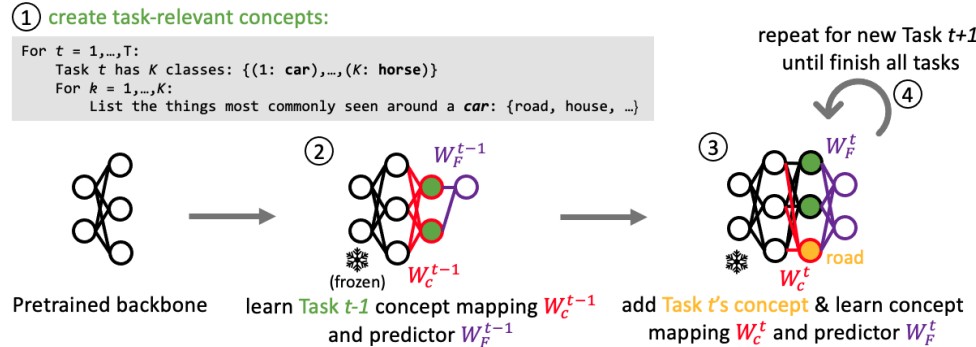

Figure 4: IN2's procedure.

## 4.2   Step 1: Concept set expansion

In this step, we expand the concept set based on classes in new task $t$. Given the concept set from $t-1$ tasks as $\mathcal{C}^{t-1}$, we form a new concept set $\mathcal{C}^t$ by adding all concepts from $c_k^t$ and $\mathcal{C}^{t-1}$. If certain concepts in $c_k^t$ are already present in $\mathcal{C}^{t-1}$, we still include them nonetheless. This is because identical textual concepts across different tasks may exhibit distinct attributes, such as variations in color and shape. For example, concept "ship" might refer to "vessel" or "cargo ship" in different tasks. After the expansion, there are $|\mathcal{C}^t| = |\mathcal{C}^{t-1}| + |c_k^t|$ concepts in $\mathcal{C}^t$. The analysis of duplicate concepts and numbers are in Appendix C.3

## 4.3   Step 2: Learning the concept mapping

To preserve learned concepts, we keep the weights of the existing concepts from the concept mapping of the previous task. Given a new concept mapping for learning the task $t$ is $W_c^t \in \mathbb{R}^{|\mathcal{C}^t| \times d}$, we first inherit the concept mapping's weights from the previous model:

$$W_c^t[i,:] = W_c^{t-1}[i-1,:] \quad \forall i \in \{1, 2, ..., |\mathcal{C}^{t-1}|\} \tag{2}$$

where $W_c^t[i,:]$ means the $i$-th row of the $W_c^t$. The second step is to learn $W_c^t$ using the procedure of CBM. Take LF-CBM Oikarinen et al. (2023) as an example. LF-CBM aims to align the concept mapping with CLIP's (Radford et al., 2021) representation of the concepts. Therefore, the procedure is to maximize the similarity between concept mapping's $f_c^t(x) = W_c^t f(x)$ and CLIP's activation matrix. The second step can be done in two ways:

1. **Finetune**: We learn $W_c^t$ on task $t$ without additional constraints. We only initialize $W_c^t$ with old concepts but these concepts will be changed on the new task.

2. **IN2**: We freeze the previous concepts' corresponding mapping in $W_c^t$ to prevent them from changing. This strategy enables us to learn the mapping of new concepts without affecting the previously learned ones. This strategy makes IN2 light and interpretable. It eliminates the need for additional memory storage for previous tasks' samples, and provides transparency regarding the retained knowledge.

## 4.4   Step 3: Learning the prediction layer

Finally, we aim to learn the prediction layer without forgetting. First, we inherit prediction weights in $W_F^{t-1}$ to $W_F^t$ before the learning process. The idea is similar to Step 2, where $|\mathcal{C}^{t-1}|$ concept output weights in $W_F^t$ inherit the same weights from $W_F^{t-1}$. Again, the **Finetune** strategy is to learn $W_F^t$ without any constraints. On the other hand, the **IN2** strategy follows the similar idea as IG-CL's Step 3 to learn the prediction layer with regularization $\gamma$ and $\beta = 2$. Meanwhile, we control the sparsity of $W_F$ by setting $\alpha = 1$ and $\lambda$. Given the dataset $\mathcal{D}_t = (\mathcal{X}^t, \mathcal{Y}^t)$, the training goal is to minimize the loss in Eq. (3) where $\mathcal{L}$ could be cross-entropy

loss:

$$\min_{\boldsymbol{\theta}^t} \mathcal{L}(W_F^t; f_c^t(\mathcal{X}^t), \mathcal{Y}^t) + \lambda\|W_F^t\|_\alpha + \gamma\|W_F^t - W_F^{t-1}\|_\beta \tag{3}$$

### 4.5 Further Discussion

**Interpretability.** IN2 is a versatile framework that supports various approaches for constructing CBMs (Koh et al., 2020; Yuksekgonul et al., 2023; Oikarinen et al., 2023), enhancing interpretability by allowing analysis of predictions in human-understandable terms, as illustrated in Figure 5. This interpretability also makes the progression of knowledge acquisition in continual learning transparent, a significant improvement over previous architecture-based methods (Yoon et al., 2018; Wang et al., 2021), which lack clarity and interpretability on the knowledge integrated into the extended architecture. Similar to IG-CL, generalizability, efficiency and accuracy are the selection criteria for CBMs. Because the CBMs for IN2 have to be accurately applied on different data streams, and be efficient for smooth continual learning processes. Based on these criteria, in this work we build IN2 upon LF-CBM (Oikarinen et al., 2023).

**Ability to Share Concepts Among Tasks.** IN2 is designed to keep the learned concepts from old tasks. Figure 5, Appendix A.2 and B.6 study IN2's ability, and show that it can improve GEM (Lopez-Paz & Ranzato, 2017) by up to 7.8% in forward transfer metric when combining with GEM.

## 5 Experiments

In Section 5.1, we introduce the datasets, evaluation metrics, baselines and other details in the experiment. In Section 5.2, we compare our methods with existing methods in standard metrics. In Section 5.3, we unveil the learned knowledge and the training process by leveraging IG-CL and IN2 interpretability. In Section 5.4, we conduct ablation studies for our methods to analyze the contributions to their superior performance. Due to page limit, additional experiments and discussions are in Appendix A, B and C. An overview of additional experiments is provided in the first page of Appendix as Appendix Outline.

### 5.1 Experiment setup

**Dataset.** To evaluate our methods, we perform experiments on two datasets: CIFAR-100 (Krizhevsky et al., 2009) and TinyImageNet (Le & Yang, 2015). Experiments on CIFAR-10 (Krizhevsky et al., 2009) and CUB-200 (Wah et al., 2011) are discussed in Appendix B and C. We consider $T = 5, 10, 20$ tasks scenario in class incremental setting. We use ResNet18 (He et al., 2016) as the experiment model. Experiment, dataset, result report and hyperparameter selection details are in Appendix D.1.

**Evaluation Metrics.** Following (Mirzadeh et al., 2022b; Chaudhry et al., 2018), we use the standard evaluation metrics to evaluate our methods. Define $a_{i,j}$ as model's accuracy on $j$-th task after learning $i$-th task, $i \geq j$. When testing the performance on $t$-th task, the metrics' definitions are as follows:

- **Average Accuracy**: Measures the average model performance $A_t = \frac{1}{t}\sum_{i=1}^{t} a_{t,i}$

- **Average Forgetting**: Measures model performance drop on previous tasks $F_t = \frac{1}{t-1}\sum_{i=1}^{t-1} \max_{j\in(1,...,t-1)}(a_{j,i} - a_{t,i})$

However, these standard metrics only reflect models' performance at the final stage. Following recent works (Zhu et al., 2022; Soutif-Cormerais et al., 2023; Zhou et al., 2023; Caccia et al., 2020; 2022; Koh et al., 2022), we also evaluate models throughout the stream. Specifically, we also report:

- **Average Incremental Accuracy**: $\bar{A}_T = \frac{1}{T-1}\sum_{t=2}^{T} A_t$

- **Average Incremental Forgetting**: $\bar{F}_T = \frac{1}{T-1}\sum_{t=2}^{T} F_t$

**Baselines.** We perform experiments on the following continual learning baselines:

- Finetune: the standard method where models are updated continuously on a series of tasks

- Exemplar-free methods

  - (regularization-based) EWC (Kirkpatrick et al., 2017), SI (Zenke et al., 2017), LwF (Li & Hoiem, 2017)
  - (architecture-based) Adam-NSCL (Wang et al., 2021), DEN (Yoon et al., 2018) in Appendix B.5.2, SSRE (Zhu et al., 2022) in Appendix B.5.1, ICICLE (Rymarczyk et al., 2023) in Appendix B.5.3

- Exemplar-based methods

  - GEM (Lopez-Paz & Ranzato, 2017) and MIR (Aljundi et al., 2019)

**Notations and Details.** For IG-CL, "IG-CL-freeze-x" means IG-CL with implementation freeze-all/ freeze-part in Step 2. For CBM based methods, "Finetune-CBM" means using **Finetune** strategy in Step 2 and Step 3, while "IN2" means using **IN2** strategy instead. "IN2-remove" means removing duplicate concepts in concept set expansion. "-GEM" and "-MIR" means combining our methods with GEM and MIR respectively. For baseline strategies, we use the implementations from a continual learning library Avalanche (Lomonaco et al., 2021). We also use Avalanche to implement our methods.

Here we discuss $T = 5$ experiment results in $\{\bar{A}_T, \bar{F}_T\}$. For different metrics $\{A_T, F_T\}$, please see Appendix B.3; for experiment results of $\{T = 10, 20\}$, please see Appendix B.4. The comparisons with SSRE (Zhu et al., 2022), DEN (Yoon et al., 2018) and ICICLE (Rymarczyk et al., 2023) are in Appendix B.5. We were not able to compare with DER (Yan et al., 2021) as their official code is incomplete. Due to scalability and efficiency, we use CLIP-Dissect (Oikarinen & Weng, 2023) as the interpretability tool for IG-CL, and LF-CBM (Oikarinen et al., 2023) for CBMs in IN2. Memory usage experiment is in Appendix C.1.

Table 2: Accuracy comparison for IG-CL. ↑ means larger values are better, while ↓ means smaller values are better. Underline means the strongest baseline, and **boldface** means the best number. In the **Ours Improvement** rows, +/- measures our best number minus the strongest baseline, and the value is in blue if our improvement is statistically significant. Our methods outperform the baselines on both $\bar{A}_T$ and $\bar{F}_T$.

| | CIFAR-100, 5T | | TinyImagenet, 5T | |
|---|---|---|---|---|
| | $\bar{A}_T \uparrow$ | $\bar{F}_T \downarrow$ | $\bar{A}_T \uparrow$ | $\bar{F}_T \downarrow$ |
| **Exemplar-free baselines** | | | | |
| Finetune | $20.47 \pm 0.67$ | $63.02 \pm 0.45$ | $16.82 \pm 2.18$ | $49.80 \pm 0.57$ |
| EWC | $\underline{20.97 \pm 0.55}$ | $62.56 \pm 0.29$ | $16.19 \pm 2.65$ | $48.42 \pm 0.39$ |
| SI | $17.75 \pm 1.37$ | $\underline{59.31 \pm 1.93}$ | $13.07 \pm 2.57$ | $\underline{44.71 \pm 2.28}$ |
| LwF | $12.74 \pm 2.15$ | $63.66 \pm 2.40$ | $16.09 \pm 3.25$ | $49.43 \pm 1.26$ |
| Adam-NSCL | $17.45 \pm 2.35$ | $59.54 \pm 3.04$ | $\underline{17.90 \pm 2.57}$ | $44.98 \pm 0.74$ |
| **Ours** | | | | |
| IG-CL-freeze-all | $\mathbf{22.37 \pm 1.20}$ | $\mathbf{58.75 \pm 0.26}$ | $\mathbf{18.19 \pm 0.76}$ | $\mathbf{43.39 \pm 0.92}$ |
| IG-CL-freeze-part | $21.73 \pm 0.79$ | $60.51 \pm 0.35$ | $18.08 \pm 0.56$ | $46.00 \pm 0.66$ |
| **Ours Improvement** | **+1.40** | +0.56 | +1.37 | +1.32 |
| **Exemplar-based baselines** | | | | |
| GEM | $23.02 \pm 1.65$ | $60.63 \pm 4.12$ | $11.29 \pm 2.62$ | $41.30 \pm 1.67$ |
| **Ours** | | | | |
| IG-CL-freeze-all-GEM | $\mathbf{28.18 \pm 2.36}$ | $\mathbf{42.61 \pm 2.12}$ | $\mathbf{12.49 \pm 1.87}$ | $\mathbf{37.78 \pm 1.04}$ |
| IG-CL-freeze-part-GEM | $25.62 \pm 2.08$ | $52.39 \pm 3.24$ | $12.06 \pm 1.58$ | $42.67 \pm 2.79$ |
| **Ours Improvement** | **+5.16** | **+18.02** | +1.20 | **+3.52** |
| **Exemplar-based baselines** | | | | |
| MIR | $27.26 \pm 0.93$ | $53.02 \pm 2.84$ | $17.81 \pm 0.82$ | $44.60 \pm 2.43$ |
| **Ours** | | | | |
| IG-CL-freeze-all-MIR | $27.04 \pm 1.08$ | $\mathbf{51.66 \pm 3.11}$ | $19.95 \pm 2.41$ | $\mathbf{43.08 \pm 0.95}$ |
| IG-CL-freeze-part-MIR | $\mathbf{27.59 \pm 0.62}$ | $52.77 \pm 2.75$ | $\mathbf{20.02 \pm 2.58}$ | $43.87 \pm 2.25$ |
| **Ours Improvement** | +0.33 | +1.36 | **+2.21** | +1.52 |

## 5.2 Quantitative Results

### 5.2.1 Comparison results for IG-CL

For IG-CL, the accuracy comparisons with existing works are in Table 2. Compared with the Exemplar-free methods, our method outperforms existing works by up to 1.4% in $\bar{A}_T$ and up to 1.3% in $\bar{F}_T$. Most of the time IG-CL performs better using freeze-all than with freeze-part. Meanwhile, the performance is comparable to or even better than Exemplar-based methods in some benchmarks. When combining IG-CL with Exemplar-based methods, both freeze-all and freeze-part tend to improve Exemplar-based method to achieve higher $\bar{A}_T$ by up to 5.1% and lower $\bar{F}_T$ by up to 18.0%. The experiment results show that IG-CL can make models more effective and forget less, either standalone or combining with other methods. Considering the experiment results of 10-task and 20-task scenario in Appendix B.4, we observe that freeze-part has better performance when task number is bigger. Overall, the freeze-all approach is suitable for a limited number of tasks and emphasizes preserving learned knowledge, while the freeze-part approach is more appropriate for scenarios involving a large number of tasks.

Table 3: Accuracy comparison for IN2. All models are pre-trained on the Place365 dataset (Zhou et al., 2017). ↑ means larger values are better, while ↓ means smaller values are better. Underline means the strongest baseline, and **boldface** means the best number. In the **Ours Improvement** rows, +/- measures our best number minus the strongest baseline, and the value is in blue if our improvement is statistically significant. Our methods clearly outperform the baselines on both $\bar{A}_T$ and $\bar{F}_T$.

|  | CIFAR-100, 5T | | TinyImagenet, 5T | |
|---|---|---|---|---|
|  | $\bar{A}_T \uparrow$ | $\bar{F}_T \downarrow$ | $\bar{A}_T \uparrow$ | $\bar{F}_T \downarrow$ |
| **Exemplar-free baselines** | | | | |
| Finetune | $22.68 \pm 0.88$ | $75.69 \pm 2.25$ | $20.50 \pm 0.45$ | $66.52 \pm 2.10$ |
| EWC | $22.20 \pm 1.56$ | $74.93 \pm 2.34$ | $19.68 \pm 1.82$ | $65.26 \pm 2.54$ |
| SI | $22.70 \pm 1.20$ | $74.44 \pm 2.45$ | $18.87 \pm 1.71$ | $62.44 \pm 0.68$ |
| LwF | $24.20 \pm 2.29$ | $74.41 \pm 3.33$ | $20.48 \pm 2.36$ | $64.17 \pm 3.08$ |
| Adam-NSCL | $23.37 \pm 2.14$ | $53.87 \pm 3.19$ | $20.17 \pm 2.04$ | $58.28 \pm 3.02$ |
| **Ours** | | | | |
| Finetune-CBM | $\mathbf{24.99 \pm 0.79}$ | $59.29 \pm 1.40$ | $21.13 \pm 1.66$ | $61.57 \pm 1.42$ |
| IN2-remove | $23.21 \pm 0.77$ | $52.41 \pm 1.37$ | $20.10 \pm 0.43$ | $52.95 \pm 1.05$ |
| IN2 | $24.25 \pm 0.86$ | $\mathbf{47.62 \pm 1.52}$ | $\mathbf{21.30 \pm 1.55}$ | $\mathbf{49.11 \pm 1.74}$ |
| **Ours Improvement** | $+0.79$ | $\mathbf{\color{blue}+6.25}$ | $+0.80$ | $\mathbf{\color{blue}+9.17}$ |
| **Exemplar-based baselines** | | | | |
| GEM | $24.27 \pm 1.57$ | $74.28 \pm 2.54$ | $10.54 \pm 0.73$ | $41.49 \pm 1.62$ |
| **Ours** | | | | |
| IN2-GEM | $\mathbf{25.41 \pm 2.06}$ | $\mathbf{66.91 \pm 2.16}$ | $\mathbf{12.14 \pm 0.53}$ | $\mathbf{38.23 \pm 0.42}$ |
| **Ours Improvement** | $+1.14$ | $\mathbf{\color{blue}+7.37}$ | $\mathbf{\color{blue}+1.60}$ | $\mathbf{\color{blue}+3.26}$ |
| **Exemplar-based baselines** | | | | |
| MIR | $24.32 \pm 2.09$ | $61.22 \pm 1.69$ | $11.58 \pm 2.08$ | $\mathbf{44.17 \pm 2.06}$ |
| **Ours** | | | | |
| IN2-MIR | $\mathbf{31.03 \pm 2.23}$ | $\mathbf{60.39 \pm 1.06}$ | $\mathbf{14.86 \pm 1.10}$ | $46.24 \pm 2.81$ |
| **Ours Improvement** | $\mathbf{\color{blue}+6.71}$ | $+0.83$ | $\mathbf{\color{blue}+3.28}$ | $-2.07$ |

### 5.2.2 Comparison results for IN2

Table 3 compares the results of IN2 and baselines. It can be seen that our Finetune-CBM already outperforms existing exemplar-free methods in balanced average accuracy $\bar{A}_T$ with often better balanced average forgetting $\bar{F}_T$, which indicates the impact of CBM in continual learning. Notably, our IN2 even has better performance on both metrics by up to 0.8% in $\bar{A}_T$ and 9.1% in $\bar{F}_T$. Additionally, combining IN2 with exemplar-based methods can further yields more improved performance for the exemplar-based methods by up to 6.7% in $\bar{A}_T$ and 7.3% in $\bar{F}_T$. However, the trade-off between interpretability and accuracy causes a minor performance drop for MIR. CBM transforms a neural network's architecture into a more interpretable one, which causes a slight decrease in accuracy as LF-CBM (Oikarinen et al., 2023) and Post-hoc CBM (Yuksekgonul et al., 2023) shows. However, IN2 increases interpretability significantly compared with the previous continual learning methods. When remove duplicate concepts in concept set expansion, both $\bar{A}_T$ and $\bar{F}_T$ are worse than original IN2, but still better than the baselines in $\bar{F}_T$ with similar $\bar{A}_T$. Since the same text concepts can represent

different visual concepts, as demonstrated in Appendix C.3, adding duplicate concepts help IN2 achieves better performance.

**Remark.** Our methods, IG-CL and IN2, introduce interpretability to continual learning for the first time. While often achieving superior performance on two standard metrics compared to baselines, it's important to emphasize that our primary goal extends beyond mere performance enhancement. Our approach may sometimes yield accuracy comparable to or slightly worse than existing methods in 5-task scenario and different settings in Appendix B, yet it stands out by significantly boosting interpretability. This innovation marks a pivotal step in making continual learning not just more effective, but also more understandable.

## 5.3    Discussion on Concept Evolution

In this section, we visualize the learned concept to give more insights on the proposed IG-CL and IN2.

**IG-CL.** For IG-CL, we study the evolution of the concepts represented by neurons as we train across different tasks. We analyze the case which we group similar classes into the same task, so we can recognize which task a concept belongs to easier. For CIFAR-100 and TinyImagenet, the class distributions are in appendix D.3. We try to maximize the diversity between tasks, which makes tasks share less concepts and become more challenging in continuous learning regime. First, we use CLIP-Dissect (Oikarinen & Weng, 2023) to analyse how many units are still detecting the same concept after learning a new task. The results are in Figure 1 and Table 4. The values are in Table 5 in Appendix A.1. Compared with existing methods, our method has better ability to retain knowledge of concepts, with by up to 62.5% improvement in the preserved ratio as Table 5 shows. We also do a case study to understand how well our method preserves the concept units from the previous tasks. Table 4 shows results for some example neurons. Some concepts are preserved after learning unrelated new tasks, which gives us insight on how our method helps avoid catastrophic forgetting. As Table 2 shows, IG-CL outperforms baselines by up to 1.3% in forgetting metric $\bar{F}_T$.

Table 4: Concept evolution for IG-CL in freeze-all implementation. We analyse the concept evolution in the layer 4 of ResNet18. The blue concept means the concept is related to the current task, while the green concept means it is unrelated. "x" stands for non-interpretable units. The results show that IG-CL can preserve the concepts from previous classes.

| | task 1 | task 2 | task 3 | task 4 | task 5 |
|---|---|---|---|---|---|
| | | | TinyImagenet, 5T | | |
| Classes | Big objects | Human-made small objects | Big animals & Natural scenes | Small animals & Sea animals | Food & Clothes & Others |
| Unit 85 | x | Kitchen | Kitchen | Kitchen | Kitchen |
| Unit 123 | x | Bedroom | Bedroom | Bedroom | Bedroom |
| Unit 129 | Bus | Bus | Bus | Bus | Bus |
| | | | CIFAR-100, 5T | | |
| | task 1 | task 2 | task 3 | task 4 | task 5 |
| Classes | Small animals & Sea animals | Natural scenes & Plants | Big animals | Big objects | Others |
| Unit 307 | x | Kitchen | Kitchen | x | x |
| Unit 310 | x | x | x | Highway | Highway |

**IN2.** For IN2, we also studied the evolution of concepts across different tasks. Figure 5 shows an example for CIFAR-100 under 5-tasks scenario, with more results in Appendix A.2. As discussed in the previous section that though our Fine-tune CBM results is already better than many Exemplary-free baselines, our IN2 is even better than Fine-tune CBM, especially on $\bar{F}_T$, with up to 12.4% improvement. This can be seen in Figure 5 that IN2 is much better at retaining and using knowledge of concepts learned from previous tasks, which is the key to help to combat against catastrophic forgetting. As Table 3 shows, IN2 outperforms baselines by up to 9.1% in forgetting metric $\bar{F}_T$.

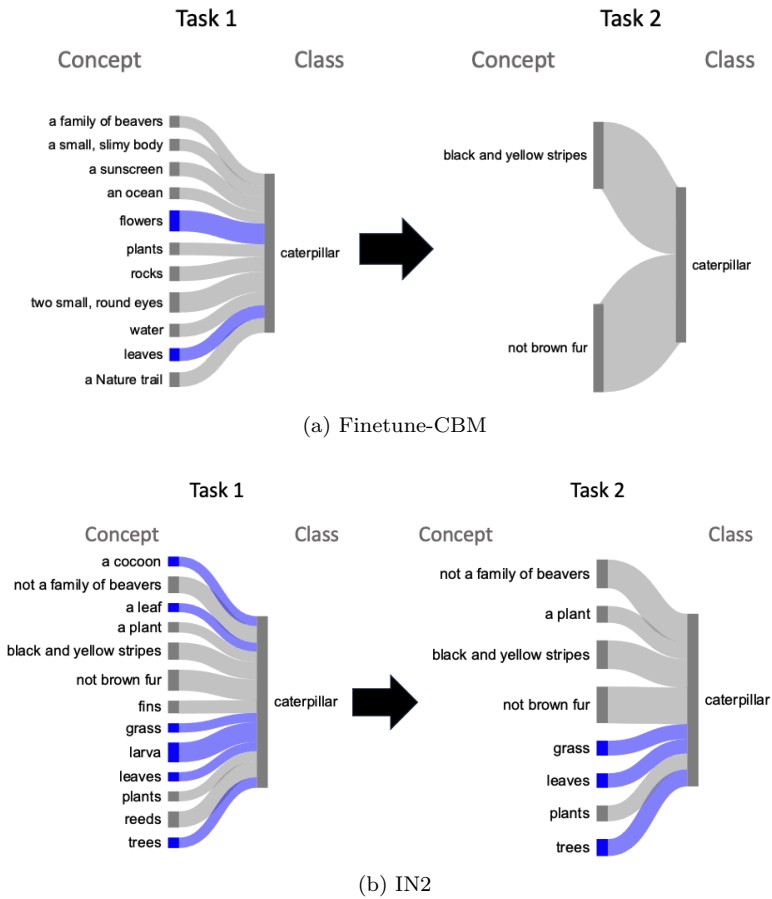

Figure 5: Final weight visualization for random classes in (a) Finetune-CBM and (b) IN2 trained on CIFAR-100 under 5-tasks scenario. For class "caterpillar"from task 1, we show its significant weight ($> 0.16$) after training on task 1 and task 2. Concepts generated from the caterpillar class itself are colored blue, and other concepts from task 1 are colored gray. The class distribution is in Table 38. We can see IN2 keeps a similar final layer while Finetune-CBM loses most significant weights.

### 5.4 Ablation Studies

To understand the contributions of IG-CL and IN2's continual learning abilities, we do ablation studies in Appendix B.1 to analysis each part of the methods. The experiment results show that utilizing all steps in Section 3 and Section 4 leads to best performances in continual learning. For IG-CL, freezing concept subnetworks improves $\bar{A}_T$ by up to 3.1% and 4.8% in $\bar{F}_T$, and regularizing $W_F$ improves $\bar{A}_T$ by up to 3.6% and 7.3% in $\bar{F}_T$. For IN2, regularizing $W_F$ improves $\bar{A}_T$ by up to 6.2% and 2.5% in $\bar{F}_T$.

## 6 Conclusion

In this work, we introduced two novel interpretable continual learning frameworks: IG-CL and IN2. The main goal of our research is to integrate interpretability into continual learning, enhancing transparency and control throughout the learning process. Although our primary focus is not on performance enhancement, our findings demonstrate that IG-CL and IN2 frequently outperform existing methods. Specifically, these frameworks improve upon previous continual learning methods by up to 1.4% in average incremental accuracy, and up to 9.1% in average incremental forgetting. Furthermore, they effectively preserve learned knowledge without the need for storing past data. The clear and interpretable nature of IG-CL and IN2 sets a strong foundation for future advancements in the field of continual learning.

## Broader Impacts

Our methods make black-box continual learning process become interpretable, which is beneficial to models' performance. However, the improvement of mitigating forgetting by controlling concepts in models might have some potential negative impact in terms of privacy. For example, if an adversary only has access to model checkpoints but not model's training data, they can analyse the concept units in the model. This will give the attacker some information about how the model was trained, and may allow them to extract some private information from the models without access to training data.

## Acknowledgement

This work is supported in part by National Science Foundation (NSF) awards CNS-1730158, ACI-1540112, ACI1541349, OAC-1826967, OAC-2112167, CNS-2100237, CNS-2120019, the University of California Office of the President, and the University of California San Diego's California Institute for Telecommunications and Information Technology/Qualcomm Institute. Thanks to CENIC for the 100Gbps networks. The authors thank the anonymous reviewers for valuable feedback on the manuscript. T. Oikarinen and T.-W. Weng are supported by National Science Foundation awards CCF-2107189, IIS-2313105, IIS-2430539. T.-W. Weng also thanks the Hellman Fellowship for providing research support.

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

## Appendix Outline

In this section we provide a brief overview of the Appendix contents.

- A:Interpretability Analysis
  - A.1: IG-CL's Concept Evolution
  - A.2: Discussion of Concept Evolution for IN2

- B: Additional Experiments
  - B.1: Ablation Study
  - B.2: Ablation on IN2 Sparsity
  - B.3: More 5 task Results
  - B.4: 10, 20 tasks Experiment Results
  - B.5: Comparison with SSRE, DEN and ICICLE
  - B.6: Foward Transfer Metric
  - B.7: Experiment Results on ImageNet-10
  - B.8: Impact from the order of tasks

- C: Further Analysis and Details
  - C.1: Computational Efficiency
  - C.2: **Interpretability Guided Continual Learning** Algorithm
  - C.3: Details of IG-CL and IN2

- D: Others
  - D.1: Experiment Setup and Details
  - D.2: Extended Previous Works Introduction
  - D.3: Class Distribution

# A  Interpretability Analysis

## A.1  IG-CL's Concept Evolution

Figure 1 and Table 4 show the IG-CL's concept evolution when grouping similar classes together. The analysis is in Section 5.3. Meanwhile, we visualize the classification head's weight for IG-CL in Figure 6. IG-CL can preserve classes' related concepts contribution to prediction. Compared with IN2 that analyzed in Figure 5, IG-CL's perserved concepts are more general and less task-specific since IG-CL can not control the emergence and types of related concepts.

Besides grouping similar classes into the same task, we also analyse the cases where class labels are distributed randomly. Following the same procedure as in section 5.3, we analyse how many units are still detecting the same concept after learning a new task. The average results for three datasets are in Table 6. Similar to Table 5, our methods outperform existing methods to retain knowledge of concepts learned from previous tasks. In general the concepts are more stable in random class distribution since different tasks might share more overlapped concepts.

Table 5: The ratio of units that still detect the same concepts they detected in the last task. Tasks' classes are in Appendix D.3. Underline means the strongest baseline, and **boldface** means the best number. Our methods outperform existing exemplar-free methods for preserving concepts, and are even better than exemplar-based methods without replaying buffer.

| Method | CIFAR100, 5T | | | | TinyImagenet, 5T | | | |
|---|---|---|---|---|---|---|---|---|
| | Task 2 | Task 3 | Task 4 | Task 5 | Task 2 | Task 3 | Task 4 | Task 5 |
| **Exemplar-free baselines** | | | | | | | | |
| Finetune | 0 | 0 | 0 | 0 | 0 | 0 | 0 | 0.086 |
| EWC | 0 | 0 | 0 | 0 | 0 | 0.030 | 0.037 | 0 |
| SI | 0 | 0 | 0 | 0 | 0.028 | 0.191 | 0.200 | 0.250 |
| LwF | 0.019 | 0.011 | 0.176 | 0 | 0.114 | 0.191 | 0.393 | 0.365 |
| **Exemplar-based baselines** | | | | | | | | |
| GEM | 0 | 0.125 | 0.076 | 0.153 | 0 | 0 | 0 | 0 |
| MIR | 0 | 0 | 0 | 0.250 | 0.028 | 0 | 0 | 0 |
| **Ours** | | | | | | | | |
| IG-CL-freeze-all (no replay) | **0.500** | **0.500** | 0.571 | **0.700** | **0.739** | **0.771** | **0.812** | **0.812** |
| IG-CL-freeze-part (no replay) | 0.125 | 0.133 | **0.727** | 0.695 | 0.515 | 0.488 | 0.544 | 0.666 |

Table 6: The ratio of units which still detect the same concepts they detected in the last task for CIFAR-10 and CIFAR-100. Class labels are distributed randomly, and the results are average over three runs. Underline means the strongest baseline, and **boldface** means the best number. Our methods outperform the existing works for preserving learned concepts.

| Method | CIFAR10, 5T | | | | CIFAR100, 5T | | | | TinyImagenet, 5T | | | |
|---|---|---|---|---|---|---|---|---|---|---|---|---|
| | Task 2 | Task 3 | Task 4 | Task 5 | Task 2 | Task 3 | Task 4 | Task 5 | Task 2 | Task 3 | Task 4 | Task 5 |
| **Exemplar-free baselines** | | | | | | | | | | | | |
| Finetune | 0 | 0 | 0 | 0 | 0.020 | 0.005 | 0.010 | 0 | 0 | 0.010 | 0 | 0.030 |
| EWC | 0 | 0 | 0 | 0 | 0.065 | 0.010 | 0 | 0 | 0.030 | 0.015 | 0 | 0.010 |
| SI | 0 | 0 | 0 | 0 | 0 | 0 | 0 | 0 | 0.011 | 0 | 0 | 0 |
| LwF | 0 | 0 | 0 | 0 | 0.020 | 0.030 | 0 | 0 | 0.015 | 0 | 0 | 0 |
| **Exemplar-based baselines** | | | | | | | | | | | | |
| GEM | 0 | 0.031 | 0 | 0 | 0.138 | 0.043 | 0.006 | 0.023 | 0 | 0.041 | 0.012 | 0.029 |
| MIR | 0 | 0 | 0 | 0 | 0.028 | 0 | 0.050 | 0.052 | 0 | 0 | 0 | 0 |
| **Ours** | | | | | | | | | | | | |
| IG-CL-freeze-all | **0.052** | **0.500** | **0.111** | **0.200** | **0.578** | **0.636** | **0.782** | 0.741 | **0.724** | **0.833** | **0.741** | 0.812 |
| IG-CL-freeze-part | 0.023 | 0.450 | 0.090 | 0.150 | 0.210 | 0.466 | 0.687 | **0.869** | 0.586 | 0.739 | 0.666 | **0.821** |

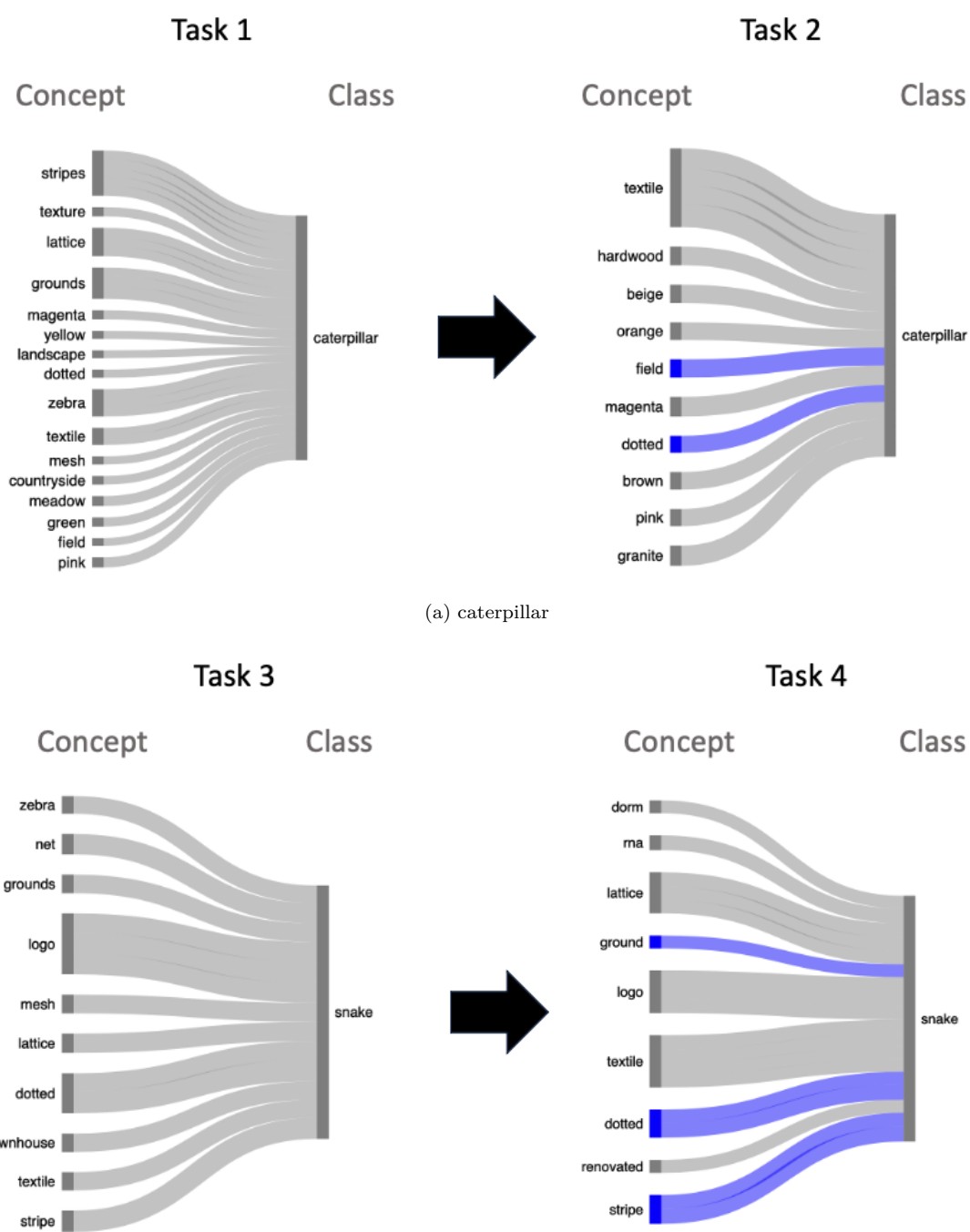

Figure 6: Final weight visualization for random classes in IG-CL on CIFAR-100 under 5-tasks scenario. For class "caterpillar" from task 1, we show its significant weight ($> 0.05$) after training on task 1 and task 2. For class "snake" from task 3, we show its significant weight after training on task 3 and task 4. Concepts related to caterpillar, and preserved from task 1 to task 2 are marked in blue. Same for concepts related to snake and preserved from task 3 to task 4. The class distribution is in Table 38. We can see IG-CL has ability to preserve related concepts' appearance and contribution.

## A.2 Discussion of Concept Evolution for IN2

In this section, we study the evolution of the concepts represented by neurons and final layer weights as we train IN2 across different tasks under 5-tasks scenario. We analyse a classes' final layer weights after learning its task, and after learning a new task. We visualize the final layer weights of Finetune-CBM and IN2 by Sankey diagrams, only including weights with absolute value greater than predefined threshold. Negative weights are reported as "NOT {concept}". The visualization for random classes in CIFAR-100 are in the main text Figure 5. The visualizations for random classes in other two datasets are in Figure 7 8. Compared with Finetune-CBM, we can see IN2 is much better at retaining and using knowledge of concepts learned from previous tasks. This helps explain why IN2 performs better in $F_T$ and $\bar{F}_T$.

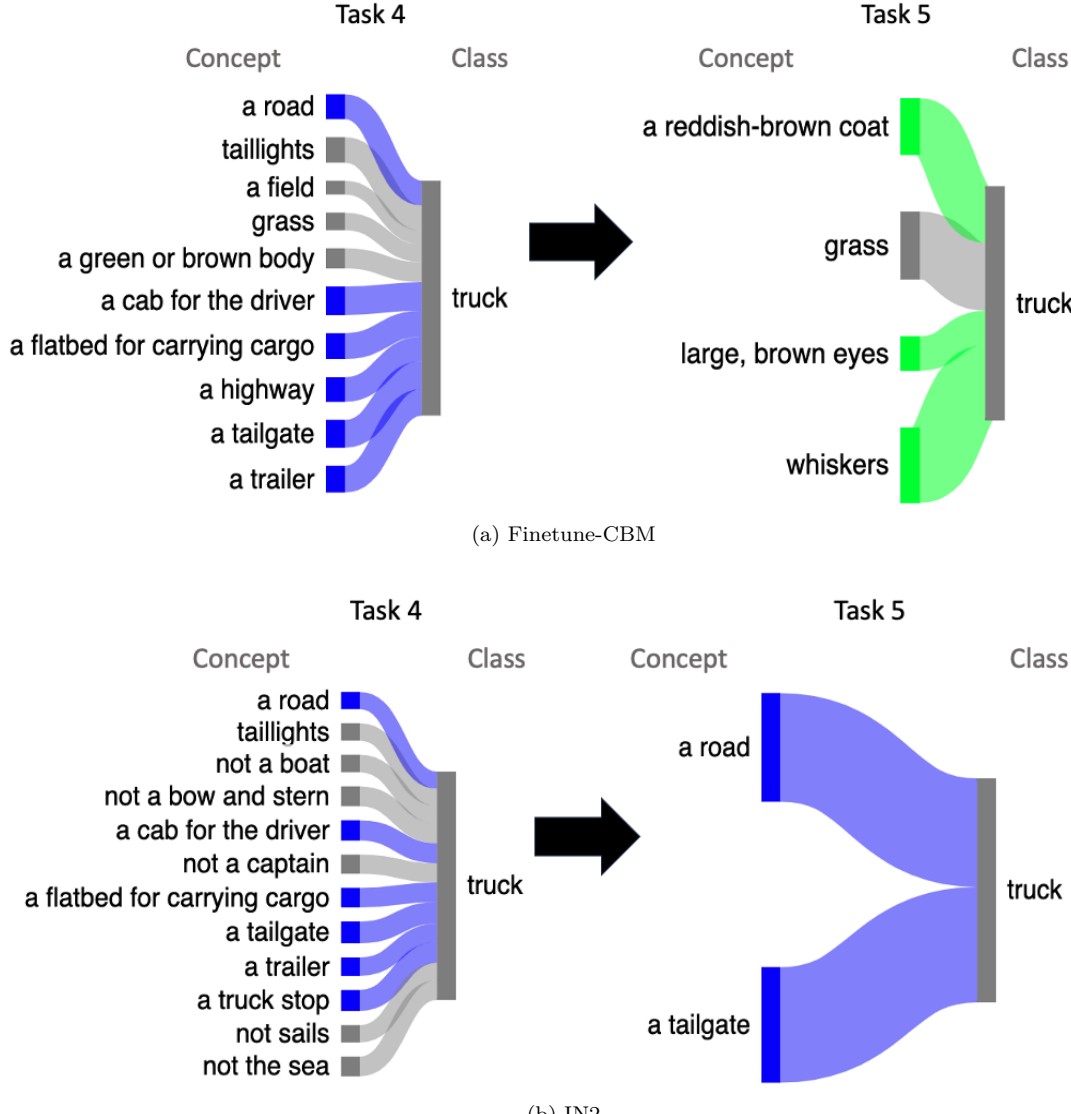

Figure 7: Final weight visualization for random classes in (a) Finetune-CBM and (b) IN2 trained on CIFAR-10 under 5-tasks scenario. For class "truck" from task 4, we show its significant weight ($> 0.15$) after training on task 4 and task 5. Concepts generated from the truck class itself are colored blue, and other concepts from task 4 are colored gray. Concepts from task 5 are colored green. The class distribution is in Table 36. We can see both models change significantly, but IN2 keeps more concepts from original task.

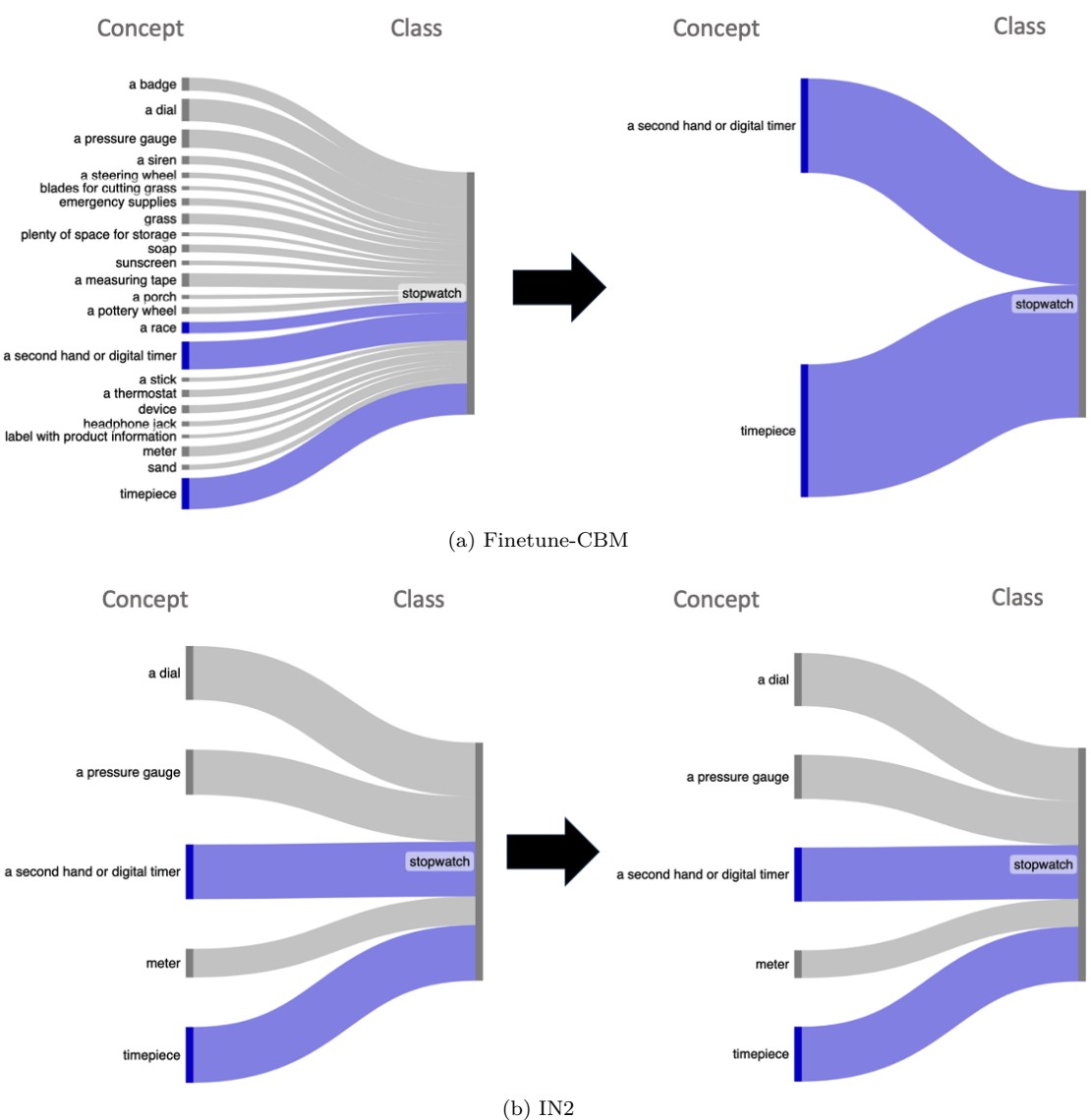

Figure 8: Final weight visualization for a random class in (a) Finetune-CBM and (b) IN2 trained on TinyImageNet under 5-tasks scenario. For class "stopwatch" from task 2, we show its significant weight ($> 0.05$) after training on task 2 and task 3. Concepts generated from the stopwatch class itself are colored blue, and other concepts from task 2 are colored gray. The class distribution is in Table 40. We can see IN2 weights stay almost identical, while Finetune-CBM changes a lot.

## B  Additional Experiments

### B.1  Ablation Study

Besides comparing the results with existing methods, we perform ablation study to analyze the impact of the key components in our methods. In Table 7, we compare IG-CL with (i) not freezing subnetwork in step 3, which is denoted as "IG-CL w/o freeze", and (ii) not regularizing parameter $\mathbf{W}_F$ in step 4, which is denoted as "IG-CL-freeze-all w/o reg" or "IG-CL-freeze-part w/o reg". The experiment results show that combining two of them results in less forgetting and better average accuracy. We believe it's because they are necessary to each other. If the concept units' subnetworks are not frozen, the concepts might change and the regularization term may not work. On the other hand, if the $\mathbf{W}_F$ changes when learning new tasks, then the concept units learned from previous tasks may not contribute to the final prediction.

For IN2, we do a similar ablation study with (i) not freezing previously learned concepts when learning CBL in step 2, which is denoted as "IN2 w/o freeze", and (ii) not regularizing parameter $\mathbf{W}_F$ in step 3, which is denoted as "IN2 w/o reg". The result Table 8 shows a similar result. In conclusion, these ablation studies demonstrate the importance of both freezing subnetworks and regularizing the $\mathbf{W}_F$ to achieving better performance and mitigating forgetting.

Table 7: Experiment results of IG-CL's ablation study. "IG-CL w/o freeze": not freezing subnetwork. "IG-CL-freeze-all w/o reg" and "IG-CL-freeze-part w/o reg": not regularizing parameter $\mathbf{W}_F$. ↑ means larger values are better, while ↓ means smaller values are better. Preserving two components turns out have the best performances.

| | CIFAR-10, 5T | | CIFAR-100, 5T | | TinyImagenet, 5T | |
|---|---|---|---|---|---|---|
| | $\bar{A}_T \uparrow$ | $\bar{F}_T \downarrow$ | $\bar{A}_T \uparrow$ | $\bar{F}_T \downarrow$ | $\bar{A}_T \uparrow$ | $\bar{F}_T \downarrow$ |
| IG-CL w/o freeze | $29.77 \pm 0.12$ | $94.76 \pm 1.48$ | $19.27 \pm 2.53$ | $60.23 \pm 6.53$ | $15.51 \pm 2.14$ | $48.26 \pm 4.57$ |
| IG-CL-freeze-all w/o reg | $30.13 \pm 0.62$ | $95.50 \pm 1.27$ | $19.32 \pm 2.02$ | $59.77 \pm 4.99$ | $14.56 \pm 2.37$ | $50.69 \pm 3.88$ |
| IG-CL-freeze-part w/o reg | $30.17 \pm 0.58$ | $95.50 \pm 1.27$ | $21.21 \pm 1.65$ | $62.04 \pm 4.58$ | $15.07 \pm 2.50$ | $51.28 \pm 4.22$ |
| IG-CL-freeze-all | $\mathbf{31.55 \pm 0.13}$ | $\mathbf{92.69 \pm 0.81}$ | $\mathbf{22.37 \pm 1.20}$ | $\mathbf{58.75 \pm 0.26}$ | $\mathbf{18.19 \pm 0.76}$ | $\mathbf{43.39 \pm 0.92}$ |
| IG-CL-freeze-part | $30.55 \pm 0.84$ | $94.47 \pm 1.12$ | $21.73 \pm 0.79$ | $60.51 \pm 0.35$ | $18.08 \pm 0.56$ | $46.00 \pm 0.66$ |

Table 8: Experiment results of IN2's ablation study. "IN2 w/o freeze": not freezing previous concepts. "IN2 w/o reg": not regularizing parameter $\mathbf{W}_F$. Preserving both has the best performance.

| | CIFAR-10, 5T | | CIFAR-100, 5T | | TinyImagenet, 5T | |
|---|---|---|---|---|---|---|
| | $\bar{A}_T \uparrow$ | $\bar{F}_T \downarrow$ | $\bar{A}_T \uparrow$ | $\bar{F}_T \downarrow$ | $\bar{A}_T \uparrow$ | $\bar{F}_T \downarrow$ |
| IN2 w/o freeze | $29.78 \pm 2.45$ | $\mathbf{87.76 \pm 1.07}$ | $18.01 \pm 3.42$ | $49.50 \pm 2.58$ | $17.30 \pm 2.89$ | $51.64 \pm 4.00$ |
| IN2 w/o reg | $28.75 \pm 1.62$ | $91.10 \pm 2.20$ | $20.42 \pm 2.68$ | $58.72 \pm 1.04$ | $19.76 \pm 2.26$ | $62.24 \pm 2.85$ |
| IN2 | $\mathbf{32.25 \pm 0.76}$ | $88.58 \pm 0.30$ | $\mathbf{24.25 \pm 0.86}$ | $\mathbf{47.62 \pm 1.52}$ | $\mathbf{21.30 \pm 1.55}$ | $\mathbf{49.11 \pm 1.74}$ |

### B.2  Ablation on IN2 Sparsity

In this section we experimented with some modifications to $W_F$ in our CBM and report their results. In our main results for IN2 and Finetune-CBM, we used a dense final layer $W_F$ instead of the sparse final layer used by (Oikarinen et al., 2023) as we found that to have best performance, and we are less focused on interpretable final decisions. Below we compare the results for CBM between dense $W_F$, and sparse $W_F$ (-S). For the sparse $W_F$, we used $\lambda = 10^{-6}$ in Eq. 3. The experiment results are in Table 9. We found that dense $W_F$ had the best performance in most cases.

Table 9: Experiment results of CBM based methods on 3 datasets. "S": sparse $W_F$. Dense $W_F$ results in best performance in average.

| | CIFAR-10,5T | | CIFAR-100, 5T | | TinyImageNet, 5T | |
|---|---|---|---|---|---|---|
| | $\bar{A}_T \uparrow$ | $\bar{F}_T \downarrow$ | $\bar{A}_T \uparrow$ | $\bar{F}_T \downarrow$ | $\bar{A}_T \uparrow$ | $\bar{F}_T \downarrow$ |
| Finetune-CBM | $30.39 \pm 0.67$ | $93.09 \pm 2.22$ | $\mathbf{24.99 \pm 0.79}$ | $59.29 \pm 1.40$ | $21.13 \pm 1.66$ | $61.57 \pm 1.42$ |
| IN2 | $\mathbf{32.25 \pm 0.76}$ | $88.58 \pm 0.30$ | $24.25 \pm 0.86$ | $47.62 \pm 1.52$ | $\mathbf{21.30 \pm 1.55}$ | $49.11 \pm 1.74$ |
| Finetune-CBM-S | $28.84 \pm 1.35$ | $91.74 \pm 2.22$ | $20.23 \pm 2.79$ | $58.29 \pm 3.12$ | $19.50 \pm 2.33$ | $61.57 \pm 2.87$ |
| IN2-S | $30.11 \pm 3.04$ | $\mathbf{87.58 \pm 0.30}$ | $19.20 \pm 3.77$ | $\mathbf{46.62 \pm 2.52}$ | $19.50 \pm 3.03$ | $\mathbf{46.11 \pm 3.48}$ |
| IN2-GEM | $\mathbf{36.37 \pm 1.74}$ | $\mathbf{44.72 \pm 2.71}$ | $\mathbf{25.41 \pm 2.06}$ | $66.91 \pm 2.16$ | $\mathbf{12.14 \pm 0.53}$ | $\mathbf{38.23 \pm 0.42}$ |
| IN2-GEM-S | $36.32 \pm 2.90$ | $45.28 \pm 4.17$ | $25.40 \pm 4.13$ | $\mathbf{65.21 \pm 3.28}$ | $5.32 \pm 1.06$ | $39.23 \pm 0.85$ |

## B.3 More 5 task Results

In Table 10 and 11, we report $A_T$ and $F_T$ for CIFAR-100 and TinyImageNet, and also experiment results for CIFAR-10 in 4 metrics.

Table 10: Accuracy comparison for IG-CL. ↑ means larger values are better, while ↓ means smaller values are better. Underline means the strongest baseline, and **boldface** means the best number. In the **Ours Improvement** rows, +/- measures our best number minus the strongest baseline, and the value is in blue if our improvement is statistically significant.

| | CIFAR-10, 5T | | CIFAR-10, 5T | | CIFAR-100, 5T | | TinyImagenet, 5T | |
|---|---|---|---|---|---|---|---|---|
| | $\bar{A}_T \uparrow$ | $\bar{F}_T \downarrow$ | $A_T \uparrow$ | $F_T \downarrow$ | $A_T \uparrow$ | $F_T \downarrow$ | $A_T \uparrow$ | $F_T \downarrow$ |
| **Exemplar-free baselines** | | | | | | | | |
| Finetune | $27.46 \pm 0.89$ | $95.86 \pm 1.28$ | $15.89 \pm 0.76$ | $78.54 \pm 0.87$ | $16.48 \pm 2.63$ | $56.65 \pm 3.24$ | $13.20 \pm 2.23$ | $43.50 \pm 0.45$ |
| EWC | $30.05 \pm 0.79$ | $\underline{94.13 \pm 2.26}$ | $18.72 \pm 1.92$ | $75.66 \pm 3.01$ | $15.79 \pm 1.58$ | $55.79 \pm 3.87$ | $13.07 \pm 3.55$ | $43.04 \pm 2.41$ |
| SI | $29.64 \pm 0.35$ | $94.21 \pm 1.66$ | $18.96 \pm 1.83$ | $\underline{74.19 \pm 1.48}$ | $13.30 \pm 2.62$ | $\mathbf{50.59 \pm 3.50}$ | $10.51 \pm 3.01$ | $\underline{39.19 \pm 3.91}$ |
| LwF | $30.16 \pm 0.23$ | $94.94 \pm 1.33$ | $19.01 \pm 0.49$ | $75.00 \pm 0.91$ | $\underline{17.92 \pm 0.10}$ | $56.85 \pm 2.54$ | $13.47 \pm 1.48$ | $43.75 \pm 1.15$ |
| Adam-NSCL | $\underline{30.23 \pm 1.02}$ | $94.82 \pm 0.53$ | $\underline{20.60 \pm 0.71}$ | $76.82 \pm 1.72$ | $17.70 \pm 0.85$ | $72.35 \pm 1.25$ | $\underline{14.15 \pm 2.19}$ | $62.71 \pm 2.93$ |
| **Ours** | | | | | | | | |
| IG-CL-freeze-all | $\mathbf{31.55 \pm 0.13}$ | $\mathbf{92.69 \pm 0.81}$ | $21.14 \pm 1.90$ | $70.68 \pm 1.72$ | $14.66 \pm 1.81$ | $51.17 \pm 2.48$ | $11.20 \pm 0.48$ | $\mathbf{37.85 \pm 0.83}$ |
| IG-CL-freeze-part | $30.55 \pm 0.84$ | $94.47 \pm 1.12$ | $20.71 \pm 2.29$ | $71.42 \pm 2.38$ | $15.17 \pm 3.47$ | $52.99 \pm 2.85$ | $11.34 \pm 3.67$ | $38.53 \pm 0.69$ |
| **Ours Improvement** | $\mathbf{+1.32}$ | $+1.44$ | $+0.54$ | $\mathbf{+3.51}$ | $-2.75$ | $-0.58$ | $-2.81$ | $+1.34$ |
| **Exemplar-based baselines** | | | | | | | | |
| GEM | $34.59 \pm 0.05$ | $\mathbf{90.00 \pm 3.80}$ | $22.51 \pm 0.22$ | $73.50 \pm 3.61$ | $19.07 \pm 0.62$ | $52.87 \pm 2.74$ | $\mathbf{11.12 \pm 1.79}$ | $37.58 \pm 1.15$ |
| **Ours** | | | | | | | | |
| IG-CL-freeze-all-GEM | $35.60 \pm 0.62$ | $94.93 \pm 0.66$ | $\mathbf{24.66 \pm 1.63}$ | $68.96 \pm 0.55$ | $\mathbf{25.68 \pm 1.72}$ | $\mathbf{36.83 \pm 0.26}$ | $9.89 \pm 2.56$ | $\mathbf{33.45 \pm 3.05}$ |
| IG-CL-freeze-part-GEM | $\mathbf{37.00 \pm 0.96}$ | $92.22 \pm 2.00$ | $23.22 \pm 1.48$ | $72.18 \pm 2.73$ | $23.70 \pm 3.84$ | $43.73 \pm 3.81$ | $10.91 \pm 2.26$ | $36.37 \pm 2.35$ |
| **Ours Improvement** | $\mathbf{+2.41}$ | $-2.22$ | $\mathbf{+2.15}$ | $\mathbf{+4.54}$ | $\mathbf{+6.61}$ | $\mathbf{+16.04}$ | $-0.21$ | $\mathbf{+4.13}$ |
| **Exemplar-based baselines** | | | | | | | | |
| MIR | $28.97 \pm 2.34$ | $\mathbf{89.80 \pm 0.99}$ | $28.18 \pm 1.73$ | $47.71 \pm 1.82$ | $9.75 \pm 3.39$ | $\mathbf{43.09 \pm 0.68}$ | $10.91 \pm 3.89$ | $38.42 \pm 1.57$ |
| **Ours** | | | | | | | | |
| IG-CL-freeze-all-MIR | $30.23 \pm 1.54$ | $91.04 \pm 2.57$ | $33.16 \pm 3.82$ | $\mathbf{42.00 \pm 2.92}$ | $14.49 \pm 0.05$ | $43.82 \pm 0.12$ | $12.29 \pm 0.60$ | $\mathbf{38.42 \pm 1.78}$ |
| IG-CL-freeze-part-MIR | $\mathbf{30.75 \pm 1.37}$ | $91.07 \pm 2.04$ | $\mathbf{34.88 \pm 1.48}$ | $42.11 \pm 2.11$ | $\mathbf{14.57 \pm 2.99}$ | $44.41 \pm 1.38$ | $\mathbf{12.47 \pm 1.02}$ | $38.50 \pm 3.76$ |
| **Ours Improvement** | $+1.78$ | $-1.24$ | $\mathbf{+6.70}$ | $\mathbf{+5.71}$ | $\mathbf{+4.82}$ | $-0.73$ | $+1.56$ | $0.00$ |

Table 11: Accuracy comparison for IN2. All models are pre-trained on the Place365 dataset (Zhou et al., 2017). ↑ means larger values are better, while ↓ means smaller values are better. Underline means the strongest baseline, and **boldface** means the best number. In the **Ours Improvement** rows, +/- measures our best number minus the strongest baseline, and the value is in blue if our improvement is statistically significant.

| | CIFAR-10, 5T | | CIFAR-10, 5T | | CIFAR-100, 5T | | TinyImagenet, 5T | |
|---|---|---|---|---|---|---|---|---|
| | $\bar{A}_T \uparrow$ | $\bar{F}_T \downarrow$ | $A_T \uparrow$ | $F_T \downarrow$ | $A_T \uparrow$ | $F_T \downarrow$ | $A_T \uparrow$ | $F_T \downarrow$ |
| **Exemplar-free baselines** | | | | | | | | |
| Finetune | $29.13 \pm 0.28$ | $97.78 \pm 0.78$ | $17.99 \pm 2.26$ | $79.66 \pm 0.72$ | $17.65 \pm 2.55$ | $64.94 \pm 2.15$ | $15.34 \pm 3.83$ | $54.81 \pm 0.71$ |
| EWC | $\underline{30.79 \pm 0.21}$ | $97.92 \pm 0.78$ | $19.00 \pm 1.87$ | $76.92 \pm 0.77$ | $17.44 \pm 3.98$ | $64.43 \pm 1.84$ | $15.07 \pm 2.23$ | $54.77 \pm 0.54$ |
| SI | $30.25 \pm 0.72$ | $\underline{96.55 \pm 1.70}$ | $19.31 \pm 0.62$ | $\underline{74.78 \pm 1.55}$ | $17.12 \pm 3.89$ | $\underline{63.11 \pm 1.03}$ | $14.47 \pm 1.02$ | $\underline{53.69 \pm 1.32}$ |
| LwF | $30.76 \pm 0.31$ | $97.75 \pm 0.84$ | $19.11 \pm 0.44$ | $76.56 \pm 1.12$ | $\underline{19.98 \pm 0.64}$ | $63.79 \pm 2.65$ | $\underline{17.08 \pm 1.50}$ | $55.40 \pm 1.85$ |
| Adam-NSCL | $30.78 \pm 0.82$ | $96.82 \pm 2.35$ | $\underline{21.60 \pm 0.48}$ | $77.73 \pm 0.86$ | $16.23 \pm 3.39$ | $66.26 \pm 3.18$ | $13.44 \pm 1.70$ | $58.86 \pm 0.70$ |
| **Ours** | | | | | | | | |
| Finetune-CBM | $30.39 \pm 0.67$ | $93.09 \pm 2.22$ | $21.14 \pm 0.83$ | $69.47 \pm 1.29$ | $16.43 \pm 2.82$ | $55.76 \pm 3.55$ | $14.91 \pm 0.65$ | $51.39 \pm 3.78$ |
| IN2 | $\mathbf{32.25 \pm 0.76}$ | $\mathbf{88.58 \pm 0.30}$ | $\mathbf{22.16 \pm 1.73}$ | $\mathbf{67.92 \pm 0.72}$ | $16.37 \pm 3.82$ | $\mathbf{45.71 \pm 2.55}$ | $15.68 \pm 1.84$ | $\mathbf{39.40 \pm 2.86}$ |
| **Ours Improvement** | $\mathbf{+1.46}$ | $\mathbf{+7.97}$ | $+0.56$ | $\mathbf{+6.86}$ | $-3.55$ | $\mathbf{+17.40}$ | $-1.40$ | $\mathbf{+14.29}$ |
| **Exemplar-based baselines** | | | | | | | | |
| GEM | $36.27 \pm 1.86$ | $69.38 \pm 2.88$ | $24.11 \pm 1.74$ | $\mathbf{59.84 \pm 1.27}$ | $20.67 \pm 1.75$ | $63.49 \pm 3.15$ | $\mathbf{5.64 \pm 0.38}$ | $37.52 \pm 0.02$ |
| **Ours** | | | | | | | | |
| IN2-GEM | $\mathbf{36.37 \pm 1.74}$ | $\mathbf{44.72 \pm 2.71}$ | $\mathbf{28.93 \pm 2.09}$ | $61.86 \pm 2.73$ | $\mathbf{23.64 \pm 3.82}$ | $\mathbf{58.54 \pm 3.75}$ | $5.37 \pm 1.67$ | $\mathbf{29.06 \pm 3.74}$ |
| **Ours Improvement** | $+0.10$ | $\mathbf{+24.66}$ | $\mathbf{+4.82}$ | $-2.02$ | $\mathbf{+2.97}$ | $-4.95$ | $-0.27$ | $\mathbf{+8.46}$ |
| **Exemplar-based baselines** | | | | | | | | |
| MIR | $33.20 \pm 2.92$ | $75.51 \pm 1.94$ | $\mathbf{33.99 \pm 2.00}$ | $48.50 \pm 2.32$ | $8.33 \pm 0.64$ | $\mathbf{46.33 \pm 2.06}$ | $2.97 \pm 3.69$ | $\mathbf{32.09 \pm 0.53}$ |
| **Ours** | | | | | | | | |
| IN2-MIR | $\mathbf{36.61 \pm 1.77}$ | $\mathbf{70.45 \pm 1.89}$ | $33.94 \pm 1.25$ | $\mathbf{38.93 \pm 1.74}$ | $\mathbf{13.81 \pm 1.94}$ | $54.76 \pm 1.59$ | $\mathbf{5.07 \pm 3.66}$ | $34.75 \pm 1.54$ |
| **Ours Improvement** | $\mathbf{+3.41}$ | $\mathbf{+5.06}$ | $-0.05$ | $\mathbf{+9.57}$ | $\mathbf{+5.48}$ | $-8.43$ | $+2.10$ | $-2.66$ |

### B.4  10, 20 tasks Experiment Results

Table 12 reports comparison for IG-CL under 10-tasks scenario for CIFAR-100 and TinyImagenet. Compared with exemplar-free baselines, IG-CL generally forgets less as $F_T$ and $\bar{F}_T$ shown. However, IG-CL sometimes is worse in $A_T$ and $\bar{A}_T$, which is the limitation of our methods. We view this as the future works to be improved, since this paper's goal is to retain intepretable concepts in models. Nevertheless, IG-CL generally improves exemplar-based baselines when combined with them, which is same as 5-tasks scenario.

Table 13 reports comparison for IN2 under 10-tasks scenario for CIFAR-100 and TinyImagenet. Similar to IG-CL, IN2 performs well in forgetting metrics $F_T$ and $\bar{F}_T$, but performs worse in $A_T$ and $\bar{A}_T$ when comparing with exemplar-free baselines. We believes this is partially because LF-CBM's training procedure is different than standard end-to-end training. As LF-CBM Oikarinen et al. (2023) paper shown, LF-CBM's classification accuracy is usually worse than standard model's. Again, we view this as future works to be improved. Besides, IN2 still can improve exemplar-based baselines when combine with them.

Table 14 and 15 report comparison for IG-CL and IN2 under 20-tasks scenario in CIFAR-100 respectively. Similar to 10-tasks scenario, our methods can improve exemplar-based baselines when combine with them, while the comparison with exemplar-free methods is still under the same trend.

We observe that both of our proposed methods can improve exemplar-based methods when combined with them. When compared with exemplar-free methods, our methods forget less while slightly worse than the strongest baselines in terms of $\bar{A}_T$. Nonetheless, we would like to highlight that our work is the first to bridge continual learning and neuron interpretability with competitive results, which opens up a new and promising direction to transform continual learning from a black-box process to more transparent.

Table 12: Accuracy comparison for IG-CL in 10 tasks scenario. ↑ means larger values are better, while ↓ means smaller values are better. Underline means the strongest baseline, and **boldface** means the best number. In the **Ours Improvement** rows, +/- measures our best number minus the strongest baseline, and the value is in blue if our improvement is statistically significant.

| | CIFAR-100, 10T | | | | TinyImagenet, 10T | | | |
|---|---|---|---|---|---|---|---|---|
| | $A_T \uparrow$ | $F_T \downarrow$ | $\bar{A}_T \uparrow$ | $\bar{F}_T \downarrow$ | $A_T \uparrow$ | $F_T \downarrow$ | $\bar{A}_T \uparrow$ | $\bar{F}_T \downarrow$ |
| **Exemplar-free baselines** | | | | | | | | |
| Finetune | $6.49 \pm 1.23$ | $65.31 \pm 1.32$ | $13.87 \pm 0.83$ | $64.20 \pm 1.52$ | $5.15 \pm 2.13$ | $49.07 \pm 0.57$ | $10.23 \pm 1.35$ | $45.83 \pm 0.21$ |
| EWC | $6.71 \pm 0.63$ | $65.32 \pm 1.12$ | $13.88 \pm 0.45$ | $63.90 \pm 0.94$ | $5.48 \pm 2.35$ | $47.42 \pm 1.47$ | $10.11 \pm 2.12$ | $45.19 \pm 1.29$ |
| SI | $6.74 \pm 1.17$ | $66.07 \pm 1.91$ | $14.01 \pm 1.37$ | $64.39 \pm 1.33$ | $5.39 \pm 1.57$ | $\underline{44.68 \pm 1.40}$ | $10.33 \pm 1.32$ | $46.21 \pm 1.36$ |
| LwF | $7.85 \pm 2.21$ | $64.13 \pm 2.19$ | $\underline{15.02 \pm 2.85}$ | $62.66 \pm 1.25$ | $5.95 \pm 2.35$ | $52.17 \pm 1.26$ | $10.82 \pm 2.35$ | $45.57 \pm 1.36$ |
| Adam-NSCL | $\underline{7.95 \pm 2.77}$ | $62.54 \pm 1.89$ | $13.69 \pm 2.15$ | $\underline{61.54 \pm 2.40}$ | $\underline{6.63 \pm 2.23}$ | $49.43 \pm 1.45$ | $\underline{10.63 \pm 1.33}$ | $\underline{49.43 \pm 1.45}$ |
| **Ours** | | | | | | | | |
| IG-CL-freeze-all | $6.99 \pm 1.20$ | $\mathbf{62.23 \pm 0.36}$ | $13.81 \pm 1.45$ | $62.79 \pm 1.70$ | $5.76 \pm 2.67$ | $48.89 \pm 2.17$ | $11.24 \pm 0.34$ | $49.39 \pm 1.92$ |
| IG-CL-freeze-part | $6.97 \pm 1.26$ | $64.03 \pm 1.32$ | $13.71 \pm 1.36$ | $63.28 \pm 1.04$ | $5.86 \pm 0.56$ | $49.99 \pm 2.16$ | $\mathbf{11.25 \pm 1.28}$ | $\mathbf{45.00 \pm 1.20}$ |
| **Ours Improvement** | -0.96 | +0.31 | -1.21 | -1.74 | -0.77 | -4.21 | +0.43 | +0.19 |
| **Exemplar-based baselines** | | | | | | | | |
| GEM | $8.25 \pm 1.07$ | $60.31 \pm 4.25$ | $15.71 \pm 1.35$ | $58.77 \pm 2.42$ | $\mathbf{6.00 \pm 2.34}$ | $47.13 \pm 1.25$ | $\mathbf{10.74 \pm 2.62}$ | $47.68 \pm 1.67$ |
| **Ours** | | | | | | | | |
| IG-CL-freeze-all-GEM | $\mathbf{15.61 \pm 1.02}$ | $\mathbf{52.01 \pm 1.34}$ | $\mathbf{29.83 \pm 2.15}$ | $\mathbf{52.43 \pm 2.36}$ | $3.03 \pm 3.05$ | $\mathbf{22.39 \pm 3.91}$ | $7.75 \pm 2.90$ | $34.75 \pm 2.37$ |
| IG-CL-freeze-part-GEM | $7.65 \pm 1.85$ | $60.12 \pm 1.37$ | $14.84 \pm 1.39$ | $54.12 \pm 2.73$ | $2.97 \pm 2.34$ | $24.91 \pm 2.02$ | $5.52 \pm 2.40$ | $\mathbf{30.44 \pm 2.59}$ |
| **Ours Improvement** | **+7.36** | **+8.30** | **+14.12** | **+6.34** | -2.97 | **+24.74** | -2.99 | **+17.24** |
| **Exemplar-based baselines** | | | | | | | | |
| MIR | $4.64 \pm 0.37$ | $59.37 \pm 1.96$ | $12.61 \pm 1.02$ | $60.51 \pm 1.72$ | $4.54 \pm 0.82$ | $46.19 \pm 2.43$ | $\mathbf{10.11 \pm 1.34}$ | $46.19 \pm 2.17$ |
| **Ours** | | | | | | | | |
| IG-CL-freeze-all-MIR | $\mathbf{5.43 \pm 1.54}$ | $52.22 \pm 1.65$ | $12.86 \pm 1.08$ | $\mathbf{56.19 \pm 2.30}$ | $5.09 \pm 3.21$ | $\mathbf{41.34 \pm 1.28}$ | $9.97 \pm 2.41$ | $\mathbf{43.21 \pm 1.32}$ |
| IG-CL-freeze-part-MIR | $5.42 \pm 1.34$ | $52.34 \pm 1.37$ | $\mathbf{12.89 \pm 0.62}$ | $56.72 \pm 1.84$ | $\mathbf{5.17 \pm 2.58}$ | $42.87 \pm 0.99$ | $9.99 \pm 1.23$ | $43.23 \pm 1.41$ |
| **Ours Improvement** | +0.79 | **+7.15** | +0.28 | **+4.32** | +0.63 | **+4.85** | -0.12 | **+2.98** |

Table 13: Accuracy comparison for IN2 in 10 tasks scenario. All models are pre-trained on the Place365 dataset (Zhou et al., 2017). ↑ means larger values are better, while ↓ means smaller values are better. Underline means the strongest baseline, and **boldface** means the best number. In the **Ours Improvement** rows, +/- measures our best number minus the strongest baseline, and the value is in blue if our improvement is statistically significant.

| | CIFAR-100, 10T | | | | TinyImagenet, 10T | | | |
|---|---|---|---|---|---|---|---|---|
| | $A_T \uparrow$ | $F_T \downarrow$ | $\bar{A}_T \uparrow$ | $\bar{F}_T \downarrow$ | $A_T \uparrow$ | $F_T \downarrow$ | $\bar{A}_T \uparrow$ | $\bar{F}_T \downarrow$ |
| **Exemplar-free baselines** | | | | | | | | |
| Finetune | $8.09 \pm 1.82$ | $80.43 \pm 1.42$ | $17.34 \pm 1.87$ | $81.23 \pm 2.30$ | $4.61 \pm 0.18$ | $\underline{66.40 \pm 3.42}$ | $14.62 \pm 0.58$ | $70.70 \pm 1.56$ |
| EWC | $7.59 \pm 1.32$ | $80.99 \pm 1.28$ | $17.23 \pm 0.96$ | $81.08 \pm 2.34$ | $5.62 \pm 2.38$ | $67.27 \pm 1.30$ | $14.88 \pm 1.28$ | $71.25 \pm 1.38$ |
| SI | $8.54 \pm 1.29$ | $81.43 \pm 1.83$ | $17.53 \pm 0.65$ | $81.60 \pm 2.45$ | $7.90 \pm 1.40$ | $71.59 \pm 0.82$ | $15.65 \pm 1.19$ | $72.33 \pm 2.08$ |
| LwF | $13.83 \pm 1.26$ | $77.21 \pm 1.53$ | $\underline{24.99 \pm 2.29}$ | $71.19 \pm 3.33$ | $\underline{10.40 \pm 2.35}$ | $71.40 \pm 1.59$ | $\underline{20.42 \pm 1.86}$ | $66.78 \pm 3.95$ |
| Adam-NSCL | $\underline{14.29 \pm 1.19}$ | $\underline{69.75 \pm 3.15}$ | $23.19 \pm 1.38$ | $76.24 \pm 1.29$ | $9.74 \pm 2.23$ | $66.87 \pm 1.45$ | $18.63 \pm 2.45$ | $69.97 \pm 1.24$ |
| **Ours** | | | | | | | | |
| Finetune-CBM | $7.31 \pm 0.82$ | $68.88 \pm 1.40$ | $14.91 \pm 1.28$ | $66.72 \pm 2.73$ | $7.18 \pm 1.26$ | $65.76 \pm 1.42$ | $14.41 \pm 1.89$ | $64.64 \pm 3.02$ |
| IN2 | $7.34 \pm 1.68$ | $\mathbf{37.90 \pm 1.59}$ | $14.98 \pm 1.09$ | $\mathbf{61.96 \pm 1.73}$ | $7.37 \pm 2.54$ | $\mathbf{56.80 \pm 1.27}$ | $14.98 \pm 2.93$ | $\mathbf{60.02 \pm 3.09}$ |
| **Ours Improvement** | -6.95 | **+31.85** | -10.01 | **+9.23** | -3.03 | **+9.60** | -5.44 | **+6.76** |
| **Exemplar-based baselines** | | | | | | | | |
| GEM | $\mathbf{11.47 \pm 1.32}$ | $66.44 \pm 3.14$ | $22.21 \pm 1.20$ | $70.54 \pm 2.32$ | $4.12 \pm 1.79$ | $54.29 \pm 1.62$ | $13.44 \pm 1.33$ | $63.58 \pm 1.11$ |
| **Ours** | | | | | | | | |
| IN2-GEM | $8.68 \pm 2.41$ | $\mathbf{53.13 \pm 3.71}$ | $\mathbf{22.46 \pm 1.15}$ | $\mathbf{59.36 \pm 2.33}$ | $\mathbf{6.20 \pm 3.07}$ | $\mathbf{47.95 \pm 1.48}$ | $\mathbf{14.96 \pm 3.48}$ | $\mathbf{57.94 \pm 1.26}$ |
| **Ours Improvement** | -2.79 | **+13.31** | +0.25 | **+11.18** | +2.08 | **+6.34** | +1.52 | **+5.64** |
| **Exemplar-based baselines** | | | | | | | | |
| MIR | $3.24 \pm 2.09$ | $\mathbf{58.77 \pm 1.69}$ | $14.19 \pm 2.21$ | $73.32 \pm 0.93$ | $3.37 \pm 0.84$ | $\mathbf{58.83 \pm 2.06}$ | $9.40 \pm 3.29$ | $\mathbf{59.29 \pm 4.20}$ |
| **Ours** | | | | | | | | |
| IN2-MIR | $\mathbf{8.38 \pm 1.35}$ | $71.02 \pm 2.06$ | $\mathbf{16.65 \pm 2.23}$ | $70.49 \pm 1.90$ | $\mathbf{7.13 \pm 2.15}$ | $60.98 \pm 2.38$ | $\mathbf{14.07 \pm 2.31}$ | $61.14 \pm 2.36$ |
| **Ours Improvement** | **+5.14** | **+12.25** | **+2.46** | **+2.83** | **+3.76** | -2.15 | **+4.67** | -1.85 |

Table 14: Accuracy comparison for IG-CL in 20 tasks scenario. ↑ means larger values are better, while ↓ means smaller values are better. Underline means the strongest baseline, and **boldface** means the best number. In the **Ours Improvement** rows, +/- measures our best number minus the strongest baseline, and the value is in blue if our improvement is statistically significant.

| | CIFAR-100, 20T | | | |
|---|---|---|---|---|
| | $A_T \uparrow$ | $F_T \downarrow$ | $\bar{A}_T \uparrow$ | $\bar{F}_T \downarrow$ |
| **Exemplar-free baselines** | | | | |
| Finetune | $3.80 \pm 2.26$ | $69.05 \pm 2.23$ | $9.72 \pm 3.54$ | $70.25 \pm 0.88$ |
| EWC | $3.56 \pm 0.16$ | $68.67 \pm 0.71$ | $9.72 \pm 2.72$ | $70.11 \pm 3.29$ |
| SI | $3.92 \pm 2.37$ | $76.58 \pm 3.77$ | $10.61 \pm 1.65$ | $76.19 \pm 0.00$ |
| LwF | $4.14 \pm 2.12$ | $77.06 \pm 1.48$ | $10.54 \pm 3.72$ | $75.25 \pm 2.74$ |
| Adam-NSCL | $\underline{10.59 \pm 0.99}$ | $67.32 \pm 3.52$ | $\underline{19.45 \pm 2.85}$ | $68.21 \pm 2.80$ |
| **Ours** | | | | |
| IG-CL-freeze-all | $3.35 \pm 1.13$ | $\mathbf{63.12 \pm 3.02}$ | $8.82 \pm 2.43$ | $65.21 \pm 2.34$ |
| IG-CL-freeze-part | $3.79 \pm 2.52$ | $72.40 \pm 3.61$ | $9.99 \pm 2.24$ | $72.52 \pm 1.42$ |
| **Ours Improvement** | -6.80 | **+4.20** | -9.46 | **+3.00** |
| **Exemplar-based baselines** | | | | |
| GEM | $14.90 \pm 3.79$ | $70.56 \pm 3.06$ | $24.88 \pm 3.34$ | $66.55 \pm 0.44$ |
| **Ours** | | | | |
| IG-CL-freeze-all-GEM | $\mathbf{20.72 \pm 0.88}$ | $\mathbf{35.48 \pm 2.14}$ | $\mathbf{27.73 \pm 2.84}$ | $66.51 \pm 1.72$ |
| IG-CL-freeze-part-GEM | $17.40 \pm 3.74$ | $63.10 \pm 2.51$ | $24.18 \pm 2.34$ | $68.03 \pm 2.30$ |
| **Ours Improvement** | **+5.82** | **+35.08** | **+2.85** | +0.04 |
| **Exemplar-based baselines** | | | | |
| MIR | $11.32 \pm 2.07$ | $45.31 \pm 0.08$ | $17.49 \pm 3.68$ | $57.71 \pm 2.67$ |
| **Ours** | | | | |
| IG-CL-freeze-all-MIR | $12.75 \pm 0.68$ | $\mathbf{39.36 \pm 2.46}$ | $17.17 \pm 2.60$ | $\mathbf{41.74 \pm 0.31}$ |
| IG-CL-freeze-part-MIR | $\mathbf{13.03 \pm 1.71}$ | $52.47 \pm 0.30$ | $\mathbf{18.02 \pm 3.30}$ | $53.93 \pm 3.61$ |
| **Ours Improvement** | +1.71 | **+5.95** | +0.53 | **+15.97** |

Table 15: Accuracy comparison for IN2 in 20 tasks scenario. All models are pre-trained on the Place365 dataset (Zhou et al., 2017). ↑ means larger values are better, while ↓ means smaller values are better. Underline means the strongest baseline, and **boldface** means the best number. In the **Ours Improvement** rows, +/- measures our best number minus the strongest baseline, and the value is in blue if our improvement is statistically significant.

| | CIFAR-100, 20T | | | |
|---|---|---|---|---|
| | $A_T \uparrow$ | $F_T \downarrow$ | $\bar{A}_T \uparrow$ | $\bar{F}_T \downarrow$ |
| **Exemplar-free baselines** | | | | |
| Finetune | $4.03 \pm 0.60$ | $71.17 \pm 0.86$ | $10.35 \pm 1.38$ | $\underline{77.07 \pm 3.09}$ |
| EWC | $3.92 \pm 3.23$ | $73.90 \pm 2.41$ | $10.65 \pm 1.20$ | $78.75 \pm 3.67$ |
| SI | $3.73 \pm 0.45$ | $78.96 \pm 0.36$ | $11.21 \pm 2.85$ | $82.57 \pm 3.67$ |
| LwF | $5.23 \pm 3.18$ | $85.31 \pm 2.62$ | $13.32 \pm 3.94$ | $85.45 \pm 3.49$ |
| Adam-NSCL | $\underline{10.93 \pm 1.33}$ | $69.52 \pm 0.54$ | $\underline{17.96 \pm 0.74}$ | $87.83 \pm 2.50$ |
| **Ours** | | | | |
| Finetune-CBM | $4.05 \pm 3.87$ | $77.40 \pm 3.48$ | $10.67 \pm 0.05$ | $74.86 \pm 3.00$ |
| IN2 | $4.04 \pm 2.87$ | $\mathbf{66.55 \pm 3.23}$ | $11.48 \pm 1.18$ | $\mathbf{73.06 \pm 0.29}$ |
| **Ours Improvement** | -6.88 | **+2.97** | -6.48 | **+4.01** |
| **Exemplar-based baselines** | | | | |
| GEM | $12.59 \pm 2.84$ | $58.33 \pm 2.69$ | $22.94 \pm 0.34$ | $\mathbf{67.12 \pm 2.08}$ |
| **Ours** | | | | |
| IN2-GEM | $\mathbf{12.87 \pm 2.37}$ | $61.74 \pm 3.69$ | $\mathbf{25.62 \pm 2.83}$ | $68.64 \pm 0.53$ |
| **Improvement** | +0.28 | -3.41 | **+2.68** | -1.52 |
| **Exemplar-based baselines** | | | | |
| MIR | $11.01 \pm 1.01$ | $29.39 \pm 2.81$ | $16.68 \pm 0.41$ | $\mathbf{51.33 \pm 1.35}$ |
| **Ours** | | | | |
| IN2-MIR | $\mathbf{12.53 \pm 0.08}$ | $36.31 \pm 1.58$ | $\mathbf{19.55 \pm 1.90}$ | $55.79 \pm 1.79$ |
| **Ours Improvement** | **+1.52** | -6.92 | **+2.87** | -4.46 |

### B.5 Comparison with More baselines

In this section, we do individual comparisons with SSRE (Zhu et al., 2022), DEN (Yoon et al., 2018) and ICICLE (Rymarczyk et al., 2023).

#### B.5.1 Comparison with SSRE

We compare our methods with SSRE (Zhu et al., 2022). Since we can not access the pre-training details described in SSRE paper, we compare our IG-CL with SSRE that is trained from scratch. Besides metrics described in Section 5.1, we also measure **Learning Accuracy** ($F_T = \frac{1}{T}\sum_{i=1}^{T} a_{i,i}$) (Mirzadeh et al., 2022b; Riemer et al., 2019; Yin et al., 2021; Mirzadeh et al., 2022a) to understand model's ability to learn new tasks. Experiment results are in Table 16. IG-CL forgets more as the performace in $F_T$ is worse. However, it has much better learning ability as $L_T$ shown.

Table 16: Comparison between IG-CL and SSRE Zhu et al. (2022) in CIFAR-10 and CIFAR-100. ↑ means larger values are better, while ↓ means smaller values are better. Even though IG-CL forgets more, it has better learning ability.

| | CIFAR-10, 5T | | | CIFAR-100, 5T | | | CIFAR-100, 10T | | | CIFAR-100, 20T | | | TinyImagenet, 5T | | | TinyImagenet, 10T | | |
|---|---|---|---|---|---|---|---|---|---|---|---|---|---|---|---|---|---|---|
| | $A_T \uparrow$ | $F_T \downarrow$ | $L_T \uparrow$ | $A_T \uparrow$ | $F_T \downarrow$ | $L_T \uparrow$ | $A_T \uparrow$ | $F_T \downarrow$ | $L_T \uparrow$ | $A_T \uparrow$ | $F_T \downarrow$ | $L_T \uparrow$ | $A_T \uparrow$ | $F_T \downarrow$ | $L_T \uparrow$ | $A_T \uparrow$ | $F_T \downarrow$ | $L_T \uparrow$ |
| SSRE | **28.44** | **15.96** | 36.18 | **31.55** | **7.97** | 32.73 | **17.72** | **7.46** | 11.29 | **8.94** | **6.73** | 4.51 | **13.76** | **6.06** | 13.59 | **11.44** | **7.33** | 15.83 |
| IG-CL-freezed-all | 21.14 | 70.68 | **93.29** | 16.66 | 51.17 | **63.22** | 6.99 | 62.23 | **64.25** | 3.35 | 63.12 | **66.46** | 13.20 | 37.85 | **47.92** | 5.76 | 48.89 | **49.81** |

#### B.5.2 Comparison with DEN

We compare our methods with DEN (Yoon et al., 2018) in the CIFAR-10 and CIFAR-100 dataset. We combine a pretrained model with a 2-layer DEN, and compare it with Finetune-CBM and IN2. The experiment results are in Table 17. IN2 and Finetune-CBM outperform DEN by up to 20.01% and 18.15% in $\bar{A}_T$ respectively.

#### B.5.3 Comparison with ICICLE

We compare IN2 with ICICLE (Rymarczyk et al., 2023) in the CUB200 (Wah et al., 2011) dataset. CUB200 is a bird species classification dataset that has 200 classes with 5994 training examples and 5794 testing examples. The experiment results are in Table 18. Our method outperforms ICICLE by 31.68% in $\bar{F}_T$.

Table 17: Comparison between DEN (Yoon et al., 2018) and our methods, in the 5-task scenario of CIFAR-10 and CIFAR-100. The experiment results shows that our methods outperform DEN in $\bar{A}_T$ by large margins.

| | CIFAR-10, 5T | | CIFAR-100, 5T | |
|---|---|---|---|---|
| | $\bar{A}_T \uparrow$ | $\bar{F}_T \downarrow$ | $\bar{A}_T \uparrow$ | $\bar{F}_T \downarrow$ |
| DEN | 12.24 | **19.20** | 9.18 | **10.50** |
| Finetune-CBM (Ours) | 30.39 | 93.09 | **24.99** | 59.29 |
| IN2 (Ours) | **32.25** | 88.58 | 24.25 | 47.62 |

Table 18: Comparison between IN2 and ICICLE (Rymarczyk et al., 2023) and other baselines in CUB200 5-task scenario. IN2 outperform ICICLE in both $\bar{A}_T$ and $\bar{F}_T$.

| | CUB200, 5T | |
|---|---|---|
| | $\bar{A}_T \uparrow$ | $\bar{F}_T \downarrow$ |
| **Exemplar-free baselines** | | |
| Finetune | 10.16 | 60.73 |
| EWC | 11.02 | 62.43 |
| LwF | 12.50 | 64.50 |
| Adam-NSCL | 13.27 | 40.73 |
| ICICLE | 11.40 | 53.66 |
| **Ours** | | |
| IN2 | 9.42 | **21.98** |

## B.6 Forward Transfer Metric

We use Forward Transfer Metric (FWT) Lopez-Paz & Ranzato (2017) to measure our methods' ability to reuse old knowledge on new tasks. We did experiments on CUB200, 5 tasks scenario since CUB200 shares several concepts among classes. Table 19 and 20 show that our methods have better abilities to share knowledge from old tasks to new tasks compared with baselines.

Table 19: Forward Transfer (FWT) Metric comparison for IG-CL. Our methods outperform the baselines on FWT by large margins.

|  | CUB200, 5T |
| --- | --- |
|  | *FWT* ↑ |
| **Exemplar-free baselines** |  |
| Finetune | 0.76 |
| EWC | 3.83 |
| SI | 13.50 |
| LwF | 10.20 |
| Adam-NSCL | 21.04 |
| **Ours** |  |
| IG-CL-freeze-all | 28.79 |
| IG-CL-freeze-part | **30.07** |
| **Exemplar-based baselines** |  |
| GEM | 1.35 |
| **Ours** |  |
| IG-CL-freeze-all-GEM | **26.76** |
| IG-CL-freeze-part-GEM | 26.68 |
| **Exemplar-based baselines** |  |
| MIR | -12.44 |
| **Ours** |  |
| IG-CL-freeze-all-MIR | 6.89 |
| IG-CL-freeze-part-MIR | **8.22** |

Table 20: Forward Transfer (FWT) Metric comparison for IN2. Our methods outperform the baselines on FWT by large margins.

|  | CUB200, 5T |
| --- | --- |
|  | *FWT* ↑ |
| **Exemplar-free baselines** |  |
| Finetune | 1.43 |
| EWC | 6.88 |
| SI | 11.67 |
| LwF | 10.36 |
| Adam-NSCL | **17.63** |
| **Ours** |  |
| Finetune-CBM | 16.71 |
| IN2 | 12.66 |
| **Exemplar-based baselines** |  |
| GEM | 19.98 |
| **Ours** |  |
| IN2-GEM | **27.78** |
| **Exemplar-based baselines** |  |
| MIR | 3.20 |
| **Ours** |  |
| IG-CL-freeze-all-MIR | **9.84** |

## B.7 Experiment results on ImageNet-10

We conduct experiments on large scale dataset ImageNet-10, and the results are in Table 21 and 22. The experiment results are still under the same trend as other datasets: Our methods outperform other exemplar-free methods by up to 0.87% in $\bar{A}_T$ and up to 5.77% in $\bar{F}_T$. Meanwhile, can improve exemplar-based methods by combining them by up to 1.83% in $\bar{A}_T$ and up to 2.87% in $\bar{F}_T$

Table 21: Experiment results on ImageNet-10 in the 5-task scenario.

|  | $\bar{A}_T$ ↑ | $\bar{F}_T$ ↓ |
| --- | --- | --- |
| Finetune | $23.33 \pm 0.75$ | $81.20 \pm 1.10$ |
| EWC | $24.14 \pm 0.69$ | $78.46 \pm 1.32$ |
| IG-CL-freeze-all (Ours) | $\mathbf{25.01} \pm 0.51$ | $\mathbf{77.20} \pm 0.95$ |
| GEM | $31.34 \pm 0.82$ | $61.71 \pm 0.82$ |
| IG-CL-freeze-all-GEM (Ours) | $\mathbf{33.17} \pm 0.50$ | $\mathbf{59.65} \pm 1.29$ |

Table 22: Experiment results on ImageNet-10 in the 5-task scenario, the backbones are pretrained on Place365 dataset.

|  | $\bar{A}_T$ ↑ | $\bar{F}_T$ ↓ |
| --- | --- | --- |
| Finetune | $23.83 \pm 0.93$ | $81.92 \pm 1.38$ |
| EWC | $\underline{25.40} \pm 0.39$ | $\underline{78.98} \pm 1.51$ |
| IN2 (Ours) | $\mathbf{26.12} \pm 0.84$ | $\mathbf{73.21} \pm 1.73$ |
| GEM | $32.66 \pm 0.85$ | $60.27 \pm 1.32$ |
| IN2-GEM (Ours) | $\mathbf{34.37} \pm 0.43$ | $\mathbf{57.40} \pm 0.99$ |

**B.8 Impact from the order of tasks**

To understand the impact from the order of tasks, we conducted experiments on CIFAR-100 using the class distribution outlined in Table 38, where similar classes are grouped into the same task. We then swapped the order of task 1 and task 2 and compared the results with the original task order. The continual learning performance for IG-CL is in Table 23, and IN2 is in Table 24. In both task orders, our methods consistently preserve learned knowledge effectively. IG-CL and IN2 outperform other exemplar-free methods by up to 0.14% in $\bar{A}_T$ and up to 26.39% in $\bar{F}_T$. Additionally, they improve upon exemplar-based methods by up to 1.45% in $\bar{A}_T$ and up to 11% in $\bar{F}_T$. These results demonstrate that our methods are robust to changes in task order. Meanwhile, it shows that our methods don't have constant performances in $\bar{A}_T$ when changing class distribution and task order, yet it boosts interpretability a lot.

We also analyze the concept evolution of two methods like Section 5.3. For IG-CL, the concept evolution analysis is in Table 25. Same as the original order that analyzed in Table 4, IG-CL can preserve concepts even after learning a unrelated task. For IN2, the analysis is in Figure 9. Same as the original order that analyzed in Figure 5, IN2 can preserve concepts while finetune CBM loss all of the significant weights. Overall, the concept evolution analysis show that our methods can preserved learned concepts regardless of the task order.

Table 23: Continual learning performance on CIFAR-100 5-task scenario. Original order is the class distribution in Table 38, and Swap order exchanges the task 1 and task 2 of it. IG-CL has better $\bar{F}_T$ and similar or better $\bar{A}_T$.

| | Original order | | Swap order | |
| --- | --- | --- | --- | --- |
| | $\bar{A}_T \uparrow$ | $\bar{F}_T \downarrow$ | $\bar{A}_T \uparrow$ | $\bar{F}_T \downarrow$ |
| Naive | 12.83 | 45.01 | 12.50 | 47.94 |
| SI | 13.10 | 41.10 | 12.87 | 47.03 |
| LwF | 14.47 | 41.83 | **14.36** | 47.00 |
| IG-CL-freeze-all | 13.93 | **36.85** | 13.63 | **43.36** |
| IG-CL-freeze-part | **14.61** | 39.19 | 13.33 | 46.29 |
| GEM | 16.57 | 40.31 | 16.01 | 45.92 |
| IG-CL-freeze-all-GEM | **17.23** | **29.31** | **16.43** | **40.41** |
| IG-CL-freeze-part-GEM | 15.88 | 34.54 | 15.08 | 41.91 |

Table 24: Continual learning performance on CIFAR-100 5-task scenario. Original order is the class distribution in Table 38, and Swap order exchanges the task 1 and task 2 of it. IN2 has better $\bar{F}_T$ and similar or better $\bar{A}_T$.

| | Original order | | Swap order | |
| --- | --- | --- | --- | --- |
| | $\bar{A}_T \uparrow$ | $\bar{F}_T \downarrow$ | $\bar{A}_T \uparrow$ | $\bar{F}_T \downarrow$ |
| Naive | 14.64 | 56.16 | 14.27 | 59.95 |
| SI | 15.20 | 55.98 | 15.58 | 59.12 |
| LwF | **18.32** | 55.26 | **18.74** | 52.13 |
| finetune_cbm | 13.95 | 37.51 | 11.41 | 42.29 |
| IN2 | 15.34 | **28.87** | 16.23 | **33.32** |
| GEM | 17.02 | 55.35 | 20.61 | 53.22 |
| IN2-GEM | **18.15** | **48.50** | **22.06** | **52.71** |

Table 25: Concept evolution for IG-CL in freeze-all implementation, testing on class distribution 38 but swap task 1 and task 2. We analyse the concept evolution in the layer 4 of ResNet18. The blue concept means the concept is related to the current task, while the green concept means it is unrelated. "x" stands for non-interpretable units. The results show that IG-CL can preserve the concepts from previous classes.

| CIFAR-100, 5T | | | | | |
|---|---|---|---|---|---|
| | task 1 | task 2 | task 3 | task 4 | task 5 |
| Classes | Natural scenes & Plants | Small animals & Sea animals | Big animals | Big objects | Others |
| Unit 275 | landscapes | landscapes | landscapes | x | x |
| Unit 140 | x | x | zebra | zebra | x |

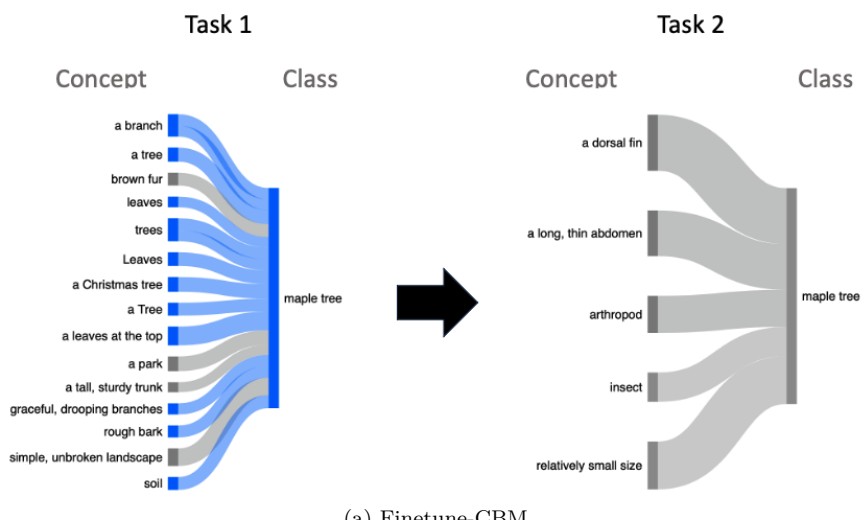

(a) Finetune-CBM

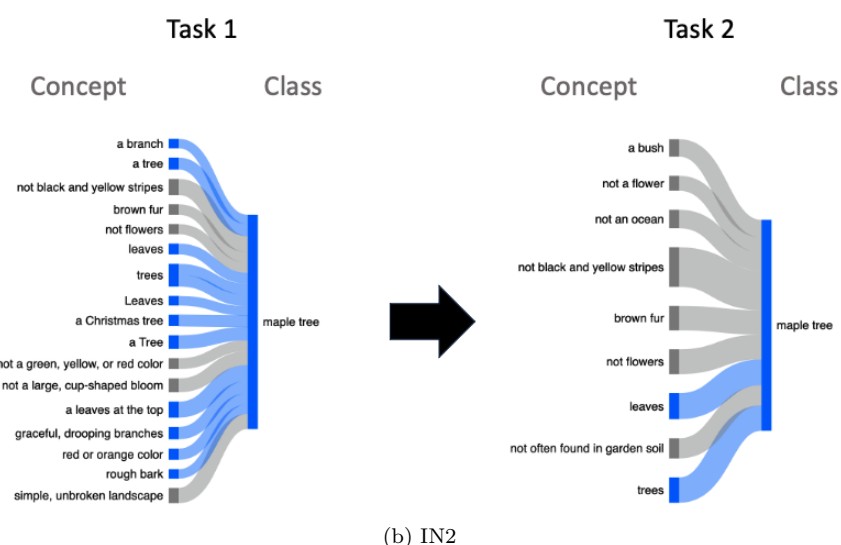

(b) IN2

Figure 9: Final weight visualization for random classes in (a) Finetune-CBM and (b) IN2 trained on CIFAR-100 under 5-tasks scenario. The class distribution is in Table 38 but swap task 1 and task 2. For class "maple tree" from task 1, we show its significant weight ($> 0.16$) after training on task 1 and task 2. Concepts generated from the maple tree class itself are colored blue, and other concepts from task 1 are colored gray. We can see IN2 keeps a similar final layer while Finetune-CBM loses most significant weights.

# C   Further Analysis and Details

## C.1   Computational efficiency

Table 26 shows the maximum GPU usage of Finetune, EWC, GEM and our methods on CIFAR-10 under 5-tasks scenario. Even integrated with the interpretable tool CLIP-Dissect (Oikarinen & Weng, 2023), IG-CL's maximum GPU usage is still lower than EWC and GEM. Meanwhile, IN2's memory usage is efficient even though the CBL layer increases when learning new tasks. The experiment results show our methods are efficient.

Table 26: Maximum GPU usage comparison for our methods, under CIFAR-10 5-tasks scenario. IG-CL's maximum GPU usage is smaller than EWC and GEM, which shows IG-CL's computational efficiency. IN2 is more efficient than existing methods.

| Method | Finetune | EWC | GEM | IG-CL-freeze-all/ IG-CL-freeze-part | IN2 |
|---|---|---|---|---|---|
| Max GPU usage (GB) | 5.51 | 5.65 | 5.68 | 5.61 | 4.00 |

## C.2   Interpretability Guided Continual Learning Algorithm

The algorithm 1 summarizes the procedure of **Interpretability Guided Continual Learning**.

---

**Algorithm 1 IG-CL**: Freeze the subnetworks of the concept units

---

**Require:** Dataset $\mathbb{D}$; regularization coefficient $\mu$; connection threshold $\tau$; regularization factor $\lambda$; Neural network parameters $\boldsymbol{\theta}$

1:  **for** $t \leftarrow 1,...,\text{T}$ **do**
2:      **if** $t$ is 1 **then**
3:          Train $\boldsymbol{\theta}^1$ on $D_t$ by solving $\min_{\boldsymbol{\theta}^1} \mathcal{L}(\boldsymbol{\theta}^1; \mathcal{D}_1) + \mu \sum_{l=1}^{L} \|\mathbf{W}_l^1\|_{2,1}$
4:      **else**
5:          Train $\boldsymbol{\theta}^t$ on $D_t$ by solving Eq. 1
6:      ConceptUnit $\leftarrow$ CLIP-Dissect($W^t$)
7:      Prev-active $\leftarrow$ ConceptUnit
8:      **for** layer $l \leftarrow \text{L},...,1$ **do**                                    ▷ Find the subnetwork of the concept units
9:          **for** Unit $u_l \leftarrow 1,...,U_l$ **do**
10:             **if** Prev-active$[u_l]$ is **True then**                              ▷ $u_l$ is in subnetwork
11:                 **for** Unit $u_{l-1} \leftarrow 1,...,U_{l-1}$ **do**
12:                     **if** $\|\mathbf{W}_{u_l,u_{l-1}}^t\|_1 > \tau$ **then**                  ▷ weight exceeds threshold
13:                         Active$[u_{l-1}] \leftarrow$ **True**
14:         **if** Using freeze-all **then**
15:             $\forall$ Prev-active$[u_l]$ is **True**, Freeze $\mathbf{W}_{u_l,:}^t$
16:             $\forall$ Active$[u_{l-1}]$ is **True**, Freeze $\mathbf{W}_{:,u_{l-1}}^t$
17:         **else if** Using freeze-part **then**
18:             $\forall$ Prev-active$[u_l]$ is **True**, Freeze $\mathbf{W}_{u_l,:}^t$
19:         Prev-active $\leftarrow$ Active

---

### C.3 Details of IG-CL and IN2

In this section, we study the technical details of the IG-CL and IN2. ImageNet-10 is the 10-class subset of ImageNet (Russakovsky et al., 2015). For IG-CL, Figure 10 shows the subnetwork sizes in different datasets. Generally, a more complicated dataset has a bigger subnetwork, but the size is adjustable by tuning hyperparameters.

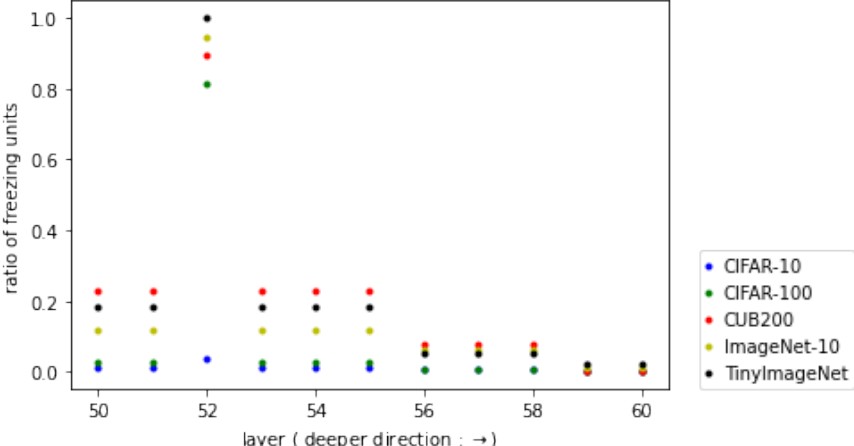

Figure 10: The subnetwork sizes in the last residual block after learning two tasks in the 5-task scenario. A more complicated dataset will have bigger subnetworks.

For IN2, Table 29 and 30 show the number of concepts added per tasks. Overall, larger and more complicated datasets have more concepts. Around 10% of concepts are the same during the concept set expansion 4.2. Table 27 shows some example concepts that are related to different classes. Table 28 shows some complex concepts in IN2 from different datasets.

| Concept | Related Classes |
|---|---|
| ship | vessel, cargo ship |
| a leaf | apple, rose |
| a small body | baby, fish |
| animal | wolf, fish, beetle |
| vehicle | bicycle, train, streetcar |

Table 27: Some concepts that generated from the different classes in IN2.

| Dataset | Complex concepts |
|---|---|
| CIFAR-10 | a flatbed for carrying cargo; four round, black tires; long hind legs for jumping |
| CIFAR-100 | a tractor trailer; a soft, round head; a crew of astronauts |
| TinyImageNet | a second hand or digital timer; a seat that is low to the ground; a double-reed mouthpiece |

Table 28: Some complex concepts that generated from the different datasets in IN2.

We build IN2 based on LF-CBM Oikarinen et al. (2023). Therefore, we would like to discuss two properties related to LF-CBM. First, following LF-CBM's procedure, we use CLIP to calculate the activation matrix $P$ as described in Section 2.2. We can replace CLIP with other vision language aligned (VL-aligned) models as long as they have a text encoder and an image encoder to calculate matrix $P$. One future work will be replacing CLIP with other suitable VL-aligned models.

Second, using GPT-3 to general concept sets is firstly proposed from an earlier work LF-CBM, which has many concurrent and follow-up works that use language models to create concept bottleneck layers (Yang et al., 2023b; Yan et al., 2023). Introducing additional information is widely exploited when designing CBM (Yuksekgonul et al., 2023; Zhou et al., 2018; Losch et al., 2019), and is reasonable task-relevant information that we can leverage. Meanwhile, using additional information to enhance a model's performance is common in other fields as well (Hu et al., 2021; Yang et al., 2023a). Even though LF-CBM has a potential leakage problem when using addtional text information, our method is still under the setting of continual learning. This is because during the training phase, the information for each class only appears in one task. One future work is to handle the potential leaking of CBM.

Table 29: Average concepts added per task in the 5-task scenario.

|  | CIFAR-10 | CIFAR-100 | CUB200 | ImageNet-10 | TinyImageNet |
|---|---|---|---|---|---|
| Number of Concept | 37.4 | 133.1 | 276.6 | 35.6 | 301.8 |

Table 30: Average concepts added per task in the 10-task scenario.

|  | CIFAR-100 | TinyImageNet |
|---|---|---|
| Number of Concept | 154.6 | 284.3 |

# D  Others

## D.1  Experiment Setup and Details

### D.1.1  Training and computing details

All models are trained on single NVIDIA V100s (32 GB SMX2). The hyperparameters used for our methods are in Table 31. We tune the hyperparameters for best performance in $\bar{A}_T$. The hyperparameters tuning results are in Table 32, 33, 34 and 35. The hyperparameter selection of $\lambda$ in Eq. (3) is discussed in Appendix B.2. The tuning process ensures that our methods are fairly compared to other baselines by optimizing their performance under the same conditions. When combine IG-CL or IN2 with GEM (Lopez-Paz & Ranzato, 2017) or MIR (Aljundi et al., 2019), we use the memory buffer size: 150 samples per task. Meanwhile, we experiment GEM and MIR with the same memory buffer size for fair comparisons. We split each dataset by 3 different random seeds, and run each class distribution for 3 times. The code and full training details will be released to public upon acceptance.

Table 31: Hyperparameters for our methods.

| $\mu$ in Eq. (1) | $10^{-6}$ |
|---|---|
| $\lambda$ in Eq. (1) | 0.4 |
| $\lambda$ in Eq. (3) | 0 |
| $\gamma$ in Eq. (3) | 0.4 |
| $\tau$ in IG-CL's Step 2 | 0.15 |

### D.1.2  Dataset details and result report

CIFAR-10/ CIFAR-100 and TinyImageNet are standard image classification benchmarks. Both CIFAR-10 and CIFAR-100 have 50k training examples and 10000 testing examples with 10 classes and 100 classes respectively. TinyImageNet has 200 classes with 500 training examples and 50 testing examples per class.

Table 32: IG-CL's performance in different $\mu$ of Eq. (1) under CIFAR-100 5-task scenario. The $\lambda$ of Eq. (1) is set to 0.4 and connection threshold $\tau$ is set to 0.15 during the experiments.

|  | $\bar{A}_T$ |
|---|---|
| $\mu = 10^{-5}$ | 20.65 |
| $\mu = 10^{-6}$ | 22.37 |
| $\mu = 10^{-7}$ | 20.19 |

Table 33: IG-CL's performance in different $\lambda$ of Eq. (1) under CIFAR-100 5-task scenario.. The $\mu$ of Eq. (1) is set to $10^{-6}$ and connection threshold $\tau$ is set to 0.15 during the experiments.

|  | $\bar{A}_T$ |
|---|---|
| $\lambda = 0.1$ | 19.28 |
| $\lambda = 0.4$ | 22.37 |
| $\lambda = 4$ | 17.17 |

Table 34: IN2's performance in different $\gamma$ of Eq. (3) under CIFAR-100 5-task scenario. The $\lambda$ of Eq. (1) is set to 0.4 and connection threshold $\tau$ is set to 0.15 during the experiments.

|  | $\bar{A}_T$ |
|---|---|
| $\gamma = 0.1$ | 22.91 |
| $\gamma = 0.4$ | 24.99 |
| $\gamma = 4$ | 18.26 |

Table 35: IG-CL's performance in different connection threshold $\tau$ under CIFAR-100 5-task scenario.. The $\mu$ and $\lambda$ of Eq. (1) is set to $10^{-6}$ and 0.4 respectively during the experiments.

|  | $\bar{A}_T$ |
|---|---|
| $\tau = 0.125$ | 19.24 |
| $\tau = 0.15$ | 22.37 |
| $\tau = 0.2$ | 17.61 |

In experiment results, we perform the independent t-test. We calculate the p-value with confidence interval 95% when comparing our best performance and the strongest baseline. The improvement is mark in blue if the p-value is less than 0.05.

## D.2 Extended Previous Works Introduction

In the following sections, we provide detailed introductions to previous continual learning methods, and the details of interpretability tools and models.

### D.2.1 Previous Continual Learning Methods

The representative method in regularization-based methods category **(i)** is **Elastic Weight Consolidation (Kirkpatrick et al., 2017)**. Elastic Weight Consolidation (EWC) is a regularized-based method. The loss function in this strategy $L(\boldsymbol{\theta}^t) = L_{\mathrm{CE}}(\boldsymbol{\theta}^t) + \frac{\lambda}{2} \sum_i F_i^t (\theta_i^t - \theta_i^{t-1})^2$ has a quadratic penalty term which is related to the difference between the parameters of the old task and the new task. Meanwhile, this penalty term is proportional to the diagonal of the Fisher information matrix F. $\lambda$ is a hyperparameter, and $\boldsymbol{\theta}^t, \boldsymbol{\theta}^{t-1}$ stand for model parameters before and after training on a new task $t$ respectively. Similarly, Zenke et al. (2017) adds a quadratic penalty term in the loss function, while estimating the importance of parameters during training. Li & Hoiem (2017) trains models with a knowledge distillation loss for old tasks and a regularization loss to outperform joint training.

The classic method in architecture-based methods category **(ii)** is **Adam-NSCL (Wang et al., 2021)**. For each task, it learns model parameters in the null space of all previous tasks, which is based on Adam. The approximate null space is obtained by using singular value decomposition to the uncentered covariance matrix of input representations of all previous tasks. Yoon et al. (2018) expands and prunes the neural network's architecture when learning new tasks.

The classic method in replay-based methods category **(iii)** is **Gradient Episodic Memory (Lopez-Paz & Ranzato, 2017)**. Gradient Episodic Memory (GEM) is a replay-based method. It stores a subset of the

observed examples from previous tasks, which is described as episodic memory. When training on new tasks, it regularizes the projection of the estimated gradient descent $g$ on the gradient descent of episodic memory $g_k$. The optimization goal is formalized as $\langle g, g_k \rangle \geq 0, \forall k < t$ when training on the task $t$. This regularization prevents the loss of episodic memory from increasing. (Rebuffi et al., 2017) stores a subset of samples for each class and uses the nearest-mean classifier on the data representation space.

### D.2.2 Detailed introduction to interpretability tools and models

CLIP-Dissect (Oikarinen & Weng, 2023) use the Contrastive Language-Image Pre-training (CLIP) model (Radford et al., 2021) to decipher the neurons in target model. CLIP is composed of an image encoder $E_I$ and a text encoder $E_T$. Given a probing dataset $\mathcal{D}_{probe} = \{x_i\}_{i=1}^N, x_i \in \mathbb{R}^d$, and a set of concepts $\{c\}_{j=1}^M$, the first step is to generate the representation by the encoders. The results are $\mathbf{I}_i = E_I(x_i), \mathbf{I}_i \in R^{I_0}$ and $\mathbf{C}_j = E_T(c_j), \mathbf{C}_j \in R^{T_0}$. Next, they compute concept-activation matrix $P \in \mathbb{R}^{N \times M}$ where $P_{i,j} = \mathbf{I}_i \cdot \mathbf{C}_j \in \mathbb{R}$. Third, neuron k's activation map is defined as $A_k(x_i)$ for input image $x_i$. The mean of activation map is described as $g(A_k(x_i)) \in \mathbb{R}$ and the activation vector is $\mathbf{q}_k = [g(A_k(x_1)), ..., g(A_k(x_N))]^T \in \mathbb{R}^N$. Finally, the concept label of neuron k is $c_n$ where $n$ is calculated from the Eq. (4)

$$n = \operatorname{argmax}_m \texttt{sim}(c_m, \mathbf{q}_k; P) \tag{4}$$

where $\texttt{sim}$ is similarity function.

Label-Free Concept Bottleneck Model (LF-CBM) (Oikarinen et al., 2023) transforms neural networks into an interpretable CBM without labeled concept data. The procedure in LF-CBM is as follows: First, it uses GPT-3 (Brown et al., 2020) to generate a set $C$ of text concepts important for the task based on class labels, which is then filtered to improve quality. Second, LF-CBM learns a CBL where each neuron corresponds to one of these concepts, by aligning the neurons with CLIP's(Radford et al., 2021) representation of the concepts. Given M concepts generated from the previous step, the CBL is a linear transformation of the pretrained NN backbone $f(x)$, expressed as $W_c \in d_0 \times |C|$. Here $d_0$ is the output dimension of $f(x)$. $W_c$ is learned to maximize the similarity between CBL's output $f_c(x) = W_c f(x)$ and CLIP's activation matrix $P$. This incentivizes the $k$-th neuron to have an activation pattern similar to CLIP with the $k$-th concept. We denote $k$-th neuron's, activation pattern as $q_k^\top = [f_{c,k}(x_1), ..., f_{c,k}(x_N)]^\top$. $W_c$ is then optimized by minimizing the following objective: $\sum_{i=1}^M -\frac{\bar{q}_i^3 \cdot \bar{P}_{:,i}^3}{\|\bar{q}_i^3\|_2 \|\bar{P}_{:,i}^3\|_2}$ where $\bar{q}$ means the vector is normalized to have mean 0 and standard derivation 1. Finally, LF-CBM learns a sparse linear prediction layer with weight $W_F$ and bias $b_F$. The optimization goal is in Eq. (5):

$$\min_{W_f^t, b_f^t} \|W_F f_c(\mathbf{X}_{\text{train}}) + b_F - y\|_2^2 + \lambda R_\alpha(W_F) \tag{5}$$

where $R_\alpha = 0.5(1-\alpha)\|W_F\|_2^2 + \alpha\|W_F\|_1$.

## D.3 Class Distribution

Table 36: Classes distribution of CIFAR-10 separated by random seed 3456.

| | |
|---|---|
| Task 1 | automobile, dog |
| Task 2 | deer, horse |
| Task 3 | bird, frog |
| Task 4 | ship, truck |
| Task 5 | airplane, cat |

Table 37: Classes distribution of CIFAR-100 separated by random seed 3456.

| | |
|---|---|
| Task 1 | bear, bee, butterfly, camel, caterpillar, chair, elephant, forest, hamster, lion, motorcycle, otter, plates, sea, shark, shrew, spider, tank, train, willow |
| Task 2 | beaver, bowls, boy, bridge, castle, cockroach, couch, dinosaur, house, keyboard, lawn mower, mushrooms, pears, pickup truck, poppies, possum, ray, skyscraper, wardrobe, whale |
| Task 3 | beetle, cloud, crocodile, lamp, leopard, lizard, palm, pine, porcupine, snail, streetcar, sweet peppers, table, telephone, television, tiger, tulips, turtle, woman, worm |
| Task 4 | baby, bed, bicycle, bottles, cans, chimpanzee, crab, lobster, man, maple, mouse, oak, orchids, plain, road, rocket, roses, skunk, squirrel, trout |
| Task 5 | apples, aquarium fish, bus, cattle, clock, cups, dolphin, flatfish, fox, girl, kangaroo, mountain, oranges, rabbit, raccoon, seal, snake, sunflowers, tractor, wolf |

Table 38: Classes distribution of CIFAR-100 separated by grouping similar classes together.

| | |
|---|---|
| Task 1 | beaver, dolphin, otter, seal, whale, aquarium fish, flatfish, ray, shark, trout, bee, beetle, butterfly, caterpillar, cockroach, crab, lobster, snail, spider, worm |
| Task 2 | maple, oak, palm, pine, willow, orchids, poppies, roses, sunflowers, tulips, apples, mushrooms, oranges, pears, sweet peppers, cloud, forest, mountain, plain, sea |
| Task 3 | hamster, mouse, rabbit, shrew, squirrel, fox, porcupine, possum, raccoon, skunk, crocodile, dinosaur, lizard, snake, turtle, bear, leopard, lion, tiger, wolf |
| Task 4 | bicycle, bus, motorcycle, pickup truck, train, lawn mower, rocket, streetcar, tank, tractor, bed, chair, couch, table, wardrobe, bridge, castle, house, road, skyscraper |
| Task 5 | baby, boy, girl, man, woman, camel, cattle, chimpanzee, elephant, kangaroo, bottles, bowls, cans, cups, plates, clock, keyboard, lamp, telephone, television |

Table 39: Classes distribution of TinyImageNet separated by random seed 3456.

| | |
|---|---|
| Task 1 | Persian cat, clIff, plunger, German shepherd, teddy, American lobster, hourglass, seashore, dumbbell, ice cream, nail, convertible, orangutan, coral reef, go-kart, king penguin, sulphur butterfly, lesser panda, kimono, comic book, cockroach, projectile, lakeside, chimpanzee, bannister, bucket, gondola, koala, lIfeboat, teapot, police van, pill bottle, hog, crane, cash machine, mushroom, water tower, black stork, ice lolly, scorpion |
| Task 2 | sewing machine, lemon, barn, Yorkshire terrier, stopwatch, lawn mower, thatch, pizza, barbershop, organ, computer keyboard, bighorn, cardigan, baboon, snail, syringe, spider web, Labrador retriever, pretzel, pomegranate, tarantula, pop bottle, trilobite, poncho, remote control, European fire salamander, altar, obelisk, binoculars, CD player, ladybug, miniskirt, cannon, wok, potter's wheel, cougar, chest, sunglasses, water jug, picket fence |
| Task 3 | rugby ball, steel arch bridge, refrigerator, espresso, dining table, monarch, brown bear, confectionery, beach wagon, scoreboard, flagpole, potpie, brass, bow tie, brain coral, backpack, chain, bison, pole, beer bottle, grasshopper, tailed frog, lion, torch, abacus, magnetic compass, standard poodle, goose, bullet train, African elephant, gazelle, triumphal arch, iPod, beacon, jinrikisha, fly, dugong, suspension bridge, ox, wooden spoon |
| Task 4 | Egyptian cat, volleyball, rocking chair, bullfrog, apron, swimming trunks, fountain, bikini, school bus, plate, guinea pig, oboe, maypole, goldfish, orange, drumstick, centipede, mashed potato, viaduct, military unIform, banana, sock, bathtub, guacamole, walking stick, pay-phone, alp, lampshade, bell pepper, meat loaf, tabby, tractor, sombrero, gasmask, frying pan, spiny lobster, jellyfish, sandal, vestment, snorkel |
| Task 5 | reel, basketball, parking meter, black widow, umbrella, trolleybus, Arabian camel, space heater, American alligator, albatross, sea cucumber, sea slug, clIff dwelling, boa constrictor, mantis, freight car, Chihuahua, fur coat, beaker, moving van, barrel, acorn, caulIflower, birdhouse, academic gown, golden retriever, neck brace, candle, desk, bee, dam, punching bag, butcher shop, slug, dragonfly, limousine, sports car, turnstile, Christmas stocking, broom |

Table 40: Classes distribution of TinyImageNet separated by by grouping similar classes together.

| | |
|---|---|
| Task 1 | school bus, maypole, projectile, freight car, pay-phone, moving van, bullet train, birdhouse, tractor, triumphal arch, cannon, police van, crane, cash machine, jinrikisha, water tower, limousine, sports car, suspension bridge, turnstile, picket fence, refrigerator, barn, lawn mower, barbershop, beach wagon, scoreboard, flagpole, trolleybus, convertible, go-kart, viaduct, pole, bathtub, altar, obelisk, bannister, gondola, lIfeboat, bucket |
| Task 2 | reel, volleyball, rocking chair, basketball, plunger, parking meter, dining table, umbrella, oboe, hourglass, computer keyboard, space heater, backpack, pop bottle, beer bottle, remote control, lampshade, torch, abacus, barrel, CD player, teapot, candle, desk, frying pan, iPod, wok, potter's wheel, pill bottle, snorkel, sunglasses, water jug, broom, wooden spoon, rugby ball, sewing machine, stopwatch, plate, teddy, drumstick |
| Task 3 | Egyptian cat, bullfrog, German shepherd, brown bear, guinea pig, Arabian camel, baboon, Labrador retriever, king penguin, lesser panda, chimpanzee, tabby, goose, koala, gazelle, golden retriever, hog, cougar, black stork, ox, Persian cat, Yorkshire terrier, bighorn, orangutan, American alligator, bison, boa constrictor, Chihuahua, lion, standard poodle, African elephant, cliff, seashore, lakeside, alp, dam, steel arch bridge, fountain, clIff dwelling, magnetic compass |
| Task 4 | monarch, snail, albatross, spider web, sulphur butterfly, tarantula, cockroach, mantis, European fire salamander, ladybug, bee, slug, dragonfly, fly, scorpion, black widow, centipede, grasshopper, tailed frog, goldfish, coral reef, sea cucumber, spiny lobster, jellyfish, American lobster, brain coral, sea slug, trilobite, lemon, banana, guacamole, mushroom, thatch, orange, mashed potato, pomegranate, bell pepper, acorn, cauliflower, binoculars |
| Task 5 | cardigan, sock, fur coat, academic gown, miniskirt, neck brace, sombrero, gasmask, punching bag, sandal, Christmas stocking, apron, swimming trunks, bikini, bow tie, military uniform, kimono, poncho, espresso, pizza, organ, potpie, ice cream, nail, pretzel, beacon, butcher shop, vestment, chest, dugong, ice lolly, confectionery, brass, comic book, meat loaf, dumbbell, syringe, chain, walking stick, beaker |

