# OpenReview forum: "Concept-Driven Continual Learning"
_TMLR — Accepted by TMLR_

### Review · Reviewer_XJSC · 2024-07-01

**Summary Of Contributions:**

This paper proposes two frameworks to solve  the challenge of catastrophic forgetting in continual learning.
These frameworks bring interpretability into continual learning, which can surpass existing methods.

**Audience:**

Yes

**Claims And Evidence:**

No

**Requested Changes:**

In this paper, authors did not clarify or explain details of designing the proposed frameworks, which make this paper hard to understand. Besides, this paper is not self-contained, such as clip-dissect, just introducing a name without introducing any details. More crucially, experimental results are not convincing enough, since authors just conduct experiments on just one model.

1. For better understanding, authors should explain how to decipher the neurons detailedly, and visualize these "concept units". Now, these detailed are missing.

2. Why only decipher the neurons in the last layer? How about decipher neurons in other layers? I suggest authors clarify this selection.

3. For the step 2 of IG-CL, I wonder when the model is trained for the task 3, whether concept units of both task 1 and task 2 are frozen?
Moreover, I think authors should describe this step 2 in mathematical equations.

4. What is the relation between the proposed IG-CL and IN2? Why not proposing a framework that can combine IG-CL and IN2?

5. How to create a  task-relevant concept set? Details are missing.

6. What is the definition of M^{t-1} and N^{t-1}? Can authors introduce their definitions?

7. Authors should explain the reason why there exists a "minus" performance in Table 3.

8 More experiments should be performed to verify the effectiveness of the proposed method, especially on more models.

**Strengths And Weaknesses:**

It is very important to consider interpretability when designing algorithms.

---

> ### Author Response · Authors · 2024-07-20
> **Author Response (1/3)**
>
> Thank you for your valuable feedback, please see our reply below to your questions. We will upload the revised draft after three reviews are available according to the review policy.
>
>  > **Q1**: For better understanding, authors should explain how to decipher the neurons detailedly, and visualize these "concept units". Now, these detailed are missing.”
>
> **A1:**
> Following your suggestions, we will include the details in Appendix D.2.2 (p. 35) of the revision on how to decipher the neurons in the revised paper, we paste the description below for your reference.
>
> To answer your question, in this work, we use the neuron-level interpretability tools [1, 2] to decipher neurons, these methods generate a description by analyzing what kinds of inputs result in high activations for the given neuron. For example, in Network Dissection [1], it deciphers a concept neuron when the overlap between the neuron activation and the predefined concept map exceeds threshold. For CLIP-Dissect [2], it calculates the similarity between the neuron activations and the pretrained CLIP matrix of text embedding and image embedding. Meanwhile, we have visualized the concept units in Table 4 (p.11) when using CLIP-Dissect as the interpretability tool in the original submission. We have added the concept units' contribution to prediction in Figure 6 (p.20).
>
> Here’s the paragraph we plan to include in the revision:
>
> *In this work, we use the neuron-level interpretability tools [1, 2] to decipher neurons, these methods generate a description by analyzing what kinds of inputs result in high activations for the given neuron. We focus on CLIP-Dissect [2] in this work due to its flexibility (open-vocabulary concepts) and efficiency (10-60x faster than Net-dissect [1]). CLIP-Dissect uses the Contrastive Language-Image Pre-training (CLIP) model to decipher the neurons in the target model. CLIP is composed of an image encoder $E_I$ and a text encoder $E_T$.*
>
> *Given a probing dataset $D_{probe}=\{x_i\}_{i=1}^N, x_i \in R^d$*, *and a set of concepts $\{c\}_{j=1}^{M}$, the first step is to generate the representation by the encoders. The results are $\mathbf{I}_i=E_I(x_i), \mathbf{I}_i \in R^{I_0}$ and  $\mathbf{C}_j=E_T(c_j), \mathbf{C}_j \in R^{T_0}$.* *Next, they compute concept-activation matrix $P \in \mathbb{R}^{N \times M}$ where $P_{i,j}=\mathbf{I}_i \cdot \mathbf{C}_j \in \mathbb{R}$.* *Third, neuron k's activation map is defined as $A_k(x_i)$ for input image $x_i$.* *The mean of activation map is described as $g(A_k(x_i)) \in \mathbb{R}$ and the activation vector is $\mathbf{q}_k=[g(A_k(x_1)),...,g(A_k(x_N))]^T \in \mathbb{R}^N$. Finally, the concept label of neuron k is $c_n$ where $n$ is calculated from the following equation: $n=\text{argmax}_m\texttt{sim}(c_m,\mathbf{q}_k;P)$ where $\texttt{sim}$ is similarity function.*
>
>
> > **Q2:** Why only decipher the neurons in the last layer? How about decipher neurons in other layers? I suggest authors clarify this selection.
>
> **A2:**
> We would like to clarify that we actually decipher the neurons in the 2nd-to-last layer. Under the representation learning setting, the models are composed by a feature extractor followed by a classification head. We decipher the last layer of the feature extractor. Our goal is to preserve the important concepts learned from the previous tasks. Previous work such as Network Dissection [1] analyzes the concept neurons in each layer, and finds out that high-level concept neurons emerge in the deep layers. Low-level concepts are basic descriptions like colors or scripts, while high-level concepts are objects or components of the images. The high-level concepts are specific to the classes in the previous tasks. Therefore, we need to preserve them. Following your suggestion, we will add this explanation to Section 3.1 (p.4) in the revision.
>
>
> > **Q3:** For the step 2 of IG-CL, I wonder when the model is trained for the task 3, whether concept units of both task 1 and task 2 are frozen? Moreover, I think authors should describe this step 2 in mathematical equations.
>
> **A3:**
> Yes, task 1 and 2’s concept units are frozen when training for task 3. We have described Step 2 mathematically in Line 7 - Line 19 of Algorithm 1 (p. 31).
>
> > **Q4:** What is the relation between the proposed IG-CL and IN2? Why not proposing a framework that can combine IG-CL and IN2?
>
> **A4:**
> As we described in the Introduction (p.2) and Section 4 (p.6), IG-CL and IN2 bridge interpretability and continual learning from two perspectives. IG-CL brings interpretability to continual learning by using external interpretability tools to guide models. On the other hand, IN2 tailor interpretable models like CBM for continual learning by leveraging their own interpretability. Therefore, IG-CL and IN2 are very different from each other, and merging them together will lose their purpose.

---

> ### Author Response · Authors · 2024-07-20
> **Author Response (2/3)**
>
> > **Q5:** How to create a task-relevant concept set? Details are missing.
>
> **A5:**
> Creation of the task-relevant concept sets depends on the Concept Bottleneck Model (CBM). For LF-CBM [3], concept sets are generated by asking GPT-3 about the concepts related to the classes. For CBM [4], concept sets are predefined and provided in the dataset. Following your suggestion, we will add this description in the Section 4.1 of the revision (p.6).
>
> > **Q6:** What is the definition of M^{t-1} and N^{t-1}? Can authors introduce their definitions?
>
> **A6:**
> $M^{t-1}$ means the number of concepts in the concept set of the CBM after learning task t-1. $N{t-1}$ means the number of concepts belonging to or generated from the task $t-1$. We discard the notation of $M^{t-1}$ and $N^{t-1}$ to make the description clearer in Section 4 (p.6 - p.7) of the revision.
>
> Here’s description we plan to include in the revised draft:
>
> *Step 1: Concept Set expansion*
>
> *In this step, we expand the concept set based on classes in new task $t$. Given the concept set from $t-1$ tasks as $\mathcal{C}^{t-1}$, we form a new concept set $\mathcal{C}^{t}$ by adding all concepts in $c^t_{k}$ into $\mathcal{C}^{t-1}$. If certain concepts in $c^t_{k}$ are already present in $\mathcal{C}^{t-1}$, we include them nonetheless. This is because identical textual concepts across different tasks may exhibit distinct attributes, such as variations in color and shape. For example, concept "ship" might refer to "vessel" or "cargo ship" in different tasks. After the expansion, there are $|C^t|=|C^{t-1}|+|c^t_k|$ concepts in $\mathcal{C}^t$.*
>
>
>
> > **Q7:** Authors should explain the reason why there exists a "minus" performance in Table 3.
>
> **A7:**
> As we discussed in Section 5.2 (p.9-10), there is a trade-off between interpretability and accuracy [3][8]. CBM transforms a neural network’s architecture into a more interpretable one, which causes a slight decrease in accuracy for LF-CBM [3], and significant decrease for Post-hoc CBM [8]. However, IN2 increases interpretability significantly compared with the previous continual learning methods. Following your suggestion, we will add this explanation to Section 5.2 (p.9-10) in the revision.
>
> > **Q8:** More experiments should be performed to verify the effectiveness of the proposed method, especially on more models.
>
> **A8:**
> Thank you for the comments! We would like to clarify that we use the same model as the previous works did [5-7] (ResNet 18) for fair comparison, and noted that they only tested on one model. Following your suggestion, we have conducted additional experiments on a different model, ResNet 34, to show the effectiveness of our proposed method. The results are in Table R1 and R2 below: It can be seen that the experiment results are still under the same trend: IG-CL and IN2 outperform other exemplar-free methods by up to 0.44% in $\bar{A}_T$ and up to 21.16% in $\bar{F}_T$. Meanwhile, it improves exemplar-based methods by up to 3.33% in $\bar{A}_T$ and up to 3.87% in $\bar{F}_T$.
>
> Table R1: IG-CL and baselines, using ResNet34 backbones under CIFAR-100 5 task scenario.
> |                       | $\bar{A_T}$ | $\bar{F_T}$ |
> | --------------------- | ---------- | ---------- |
> | Naive                 | 16.57      | 57.72      |
> | SI                    | 17.40      | 55.46      |
> | LwF                   | **18.98**      | 56.96      |
> | IG-CL-freeze-all      | 17.96      | 54.35      |
> | IG-CL-freeze-part     | 17.97      | **54.33**      |
> | GEM                   | 19.79      | 50.47      |
> | IG-CL-freeze-all-GEM  | **23.12**      | **46.60**  |
> | IG-CL-freeze-part-GEM | 22.30      | 47.31      |
> | MIR                   | 17.41      | 51.62      |
> | IG-CL-freeze-all-MIR  | **18.16**      | **50.51**      |
> | IG-CL-freeze-part-MIR | 18.92      | 52.17      |
>
> Table R2: IN2 and baselines, using ResNet34 backbones under CIFAR-100 5 task scenario.
> |              | $\bar{A_T}$ | $\bar{F_T}$ |
> | ------------ | ---------- | ---------- |
> | Naive        | 20.99      | 67.95      |
> | SI           | 21.24      | 67.79      |
> | LwF          | 21.32      | 66.31      |
> | Finetune CBM | 19.46      | 59.58      |
> | IN2          | **21.76**      | **45.15**      |
> | GEM          | 24.28      | **54.65**      |
> | IN2-GEM      | **24.76**      | 56.59      |
> | MIR          | 19.03      | 64.02      |
> | IN2-MIR      | **19.18**      | **60.45**      |

---

> > ### Author Response · Authors · 2024-07-20
> > **Author Response (3/3)**
> >
> > **Summary**
> >
> > To summarize, we have:
> > * Clarified in **A1** for the mechanism of deciphering neurons.
> > * Clarified in **A2** for why only decipher the last layer of feature extractor.
> > * Described the IG-CL step 2 in **A3** mathematically.
> > * Clarified the relation between IG-CL and IN2 in **A4**.
> > * Described the creation of task-relevant concepts for IN2 in **A5**.
> > * Improved the notation of IN2’s concept set in **A6**.
> > * Explained the minus performance in Table 3 in **A7**.
> > * Experimented on another model in **A8**.
> >
> > We believe we have addressed all your concerns. Please let us know if you still have any reservations and we would be happy to address them.
> >
> > **Reference:**
> >
> > [1] Understanding the role of individual units in a deep neural network. PNAS 2020
> >
> > [2] CLIP-Dissect: Automatic Description of Neuron Representations in Deep Vision Networks. ICLR 2023
> >
> > [3] Label-free Concept Bottleneck Models. ICLR 2023
> >
> > [4] Concept Bottleneck Models. ICML 2020
> >
> > [5] Self-sustaining representation expansion for non- exemplar class-incremental learning. CVPR 2022
> >
> > [6] Training networks in null space of feature covariance for continual learning. CVPR 2021
> >
> > [7] Icicle: Interpretable class incremental continual learning. ICCV 2023
> >
> > [8] Post-hoc Concept Bottleneck Models. ICLR 2023

---

> > > ### Author Response · Authors · 2024-07-27
> > > **Update revised paper and response**
> > >
> > > Dear Reviewer XJSC,
> > >
> > > We have updated the revised draft based on your suggestion. The revisions based on your suggestions are marked in blue.  Meanwhile, we have updated the table and figure numbers in the original response.
> > >
> > > We believe we have addressed all your concerns. Please let us know if you still have any reservations and we would be happy to address them.

---

### Review · Reviewer_9mSG · 2024-07-08

**Summary Of Contributions:**

This paper proposes two methods (namely IG-CL and IN2) to imbue interpretability to the continual learning task. IG-CL finds the concept units in the network and freezes them to preserve the initial knowledge, while IN2 uses a "concept bottleneck model" approach to inject the knowledge from each task to predetermined regions in the network. Both methods seem reasonable and they show strong numbers.

**Audience:**

Yes

**Broader Impact Concerns:**

The paper addresses well enough potential privacy ethics violations

**Claims And Evidence:**

Yes

**Requested Changes:**

Please consider removing the overstatement

**Strengths And Weaknesses:**

Strengths

Interpretability is a property that should always be valued. This paper addresses the problem of continual learning, which is a very useful task, in an interpretabe manner. In my opinion both the proposed methods are needed in this paper: IG-CL is useful for models that were not trained initially for continual learning, while IN2 has this knowledge from the begining and it strategizes the continual learning process. Both consepts are useful and the paper shows good performance in both cases.

Weaknesses

1) Minor syntactical errors like "fall into below main categories:" in Introduction.

2) The column "Flexibility - For any dataset" in table 1 seems obsolete given that every method is noted as "yes"

3) Overstatement: "IG-CL and IN2 represent significant advancements in continual learning" page 2 last paragraph. For something to be a significant advancement it must be judged by the scientific community and to pass the test of time

---

> ### Author Response · Authors · 2024-07-27
> **Author Response**
>
> Thank you for your valuable and positive feedback! We have revised our draft based on your suggestion in the following:
>
>
> * Fixed syntactical error in Introduction (p.1).
> * Removed the column "Flexibility - For any dataset" in Table 1 (p.3).
> * Removed overstatement in Introduction (p.2).
>
> The modification is marked in orange. Please let us know if you have other questions and we would be happy to discuss further.

---

### Review · Reviewer_8Gdw · 2024-07-20

**Summary Of Contributions:**

The authors aimed to incorporate interpretability into continual learning while mitigating catastrophic forgetting, two methods were proposed. The first method employs an external tool to decipher neuron activations, subsequently freezing them within a regularization-based framework. The second method adapts the concept bottleneck model (CBM), expanding existing concepts into new ones, and either fine-tuning all model weights or selectively freezing previous concept weights to learn those associated with the new concepts. The authors evaluated their approach on CIFAR-100 and Tiny Images for image classification tasks, reporting modest accuracy improvements over standard continual learning methods.

**Audience:**

Yes

**Broader Impact Concerns:**

Controlling concepts in models might have some potential negative impact in privacy, which might be used in adversary attacks.

**Claims And Evidence:**

Yes

**Requested Changes:**

### Enhanced Clarification of Proposed Methods:
* To better clarify the relationship and differences between the proposed methods, the authors should provide more detailed explanations.
* For instance, adding similar weight visualization for the first method would enable clearer comparisons between concept units and CBM concept sets.

### Comprehensive Experiments for Effectiveness Validation:
* While validating effectiveness with an image classification task may suffice for preliminary exploration, more comprehensive experiments are required to convince readers of the proposed concept-driven continual learning approach.
* These experiments should encompass a wider range of tasks and include a deep analysis of important variables, such as the order of tasks presented in continual pre-training.

**Strengths And Weaknesses:**

### Strengths
* Improving the interpretability of continual training is important. The paper provides some interesting discussions by identifying concept units or reusing the concept sets constructed by CBM to preserve learned knowledge and increase interpretability in continual learning.
* Technically correct. The author reported some gains on two image classification sets.
* Overall the paper is written clearly.

### Weaknesses:
* The first method does not suggest globally stable strategies (freeze-part or freeze-all), as shown in Table 2.
* The second method's assumption that adding all new concepts by treating instances from the same concept, albeit with different attributes, is the optimal approach is questionable. There's an opportunity to consider mappings to previously learned concepts when integrating new concepts. This would enable continual learning to better leverage learned knowledge and reduce redundancy in concepts.
* The paper is more a proof-of-concept and lacks convincing evidence for the general effectiveness of the proposed changes. The limited number of experimented tasks hinders a comprehensive evaluation. The authors should consider experimenting with more tasks and conducting further discussions, such as ablation studies for the order of tasks in continual training. Moreover, it's unclear how the proposed methods would handle complex compound concepts.

---

> ### Author Response · Authors · 2024-07-27
> **Author Response (1/2)**
>
> Thank you for your valuable feedback, please see our reply below to your questions. The revisions from your suggestions are marked in red in the manuscript.
>
> > **Q1:** The first method does not suggest globally stable strategies (freeze-part or freeze-all), as shown in Table 2.
>
> **A1:**
> Overall, the freeze-all approach is suitable for a limited number of tasks and emphasizes preserving learned knowledge, while the freeze-part approach is more appropriate for scenarios involving a large number of tasks. The freeze-all method is better at preserving concepts, resulting in better average forgetting performance. However, freeze-all also sacrifices the model's ability to learn new tasks, which leads to higher average accuracy for the freeze-part approach in scenarios with many tasks, such as 20 tasks in Appendix B.4 (p.25). We have added this description to Section 5.2 in the revision (p.9).
>
> > **Q2:** The second method's assumption that adding all new concepts by treating instances from the same concept, albeit with different attributes, is the optimal approach is questionable. There's an opportunity to consider mappings to previously learned concepts when integrating new concepts. This would enable continual learning to better leverage learned knowledge and reduce redundancy in concepts.
>
> **A2:**
> First, we would like to clarify that our approach preserves the mapping from previously learned concepts when incorporating new text concepts with the same name. This enables the models to leverage existing knowledge effectively. We found that around 10% of concepts are duplicate when learning a new task. We give some examples of same text concepts related in different classes in Table 27 of Appendix C.3 in the revision (p. 32) Following your suggestion, we conduct additional experiments to remove duplicate concepts, and the results are in Table 3 (p. 10). Compared with the original IN2, removing duplicate concepts results in worse $\bar{A}_T$ and $\bar{F}_T$. However, it is still better or similar than the strongest baselines, which shows our contribution. Adding these same “text” concepts that represent different “visual” concepts help IN2 achieve better performance as we explained in Section 5.2.2 of the revision (p.10). Meanwhile, we would like to highlight that IG-CL won’t have this problem since the concepts are not added manually.
>
> > **Q3:** The limited number of experimented tasks hinders a comprehensive evaluation. The authors should consider experimenting with more tasks.
>
> **A3:**
> We have done experiments with 10-task and 20-task scenarios in the original submission. The results are in appendix B.4 (p.25).  We observe that both of our proposed methods can improve exemplar-based methods when combined with them. When compared with exemplar-free methods, our methods forget less while slightly worse than the strongest baselines in terms of $\bar{A}_T$. Nonetheless, we would like to highlight that our work is the first to bridge continual learning and neuron interpretability with competitive results, which opens up a new and promising direction to transform continual learning from a black-box process to more transparent.
>
> > **Q4:** … and conducting further discussions, such as ablation studies for the order of tasks in continual training.
>
> > … more comprehensive experiments are required to convince readers of the proposed concept-driven continual learning approach. These experiments should encompass a wider range of tasks and include a deep analysis of important variables, such as the order of tasks presented in continual pre-training.
>
> **A4:** Following your suggestion, we have added experiments on CIFAR-100 using the class distribution outlined in Table 39 at p.37, where similar classes are grouped into the same task. We then swapped the order of task 1 and task 2 and compared the results with the original task order, as analyzed in Section 5.3. The additional experiments are in Appendix B.8 of the revision (p.29-30). The experimental results demonstrate that our methods are robust to changes in task order, as evidenced by continual learning performance and concept evolution analysis.

---

> ### Author Response · Authors · 2024-07-27
> **Author Response (2/2)**
>
> > **Q5:** it's unclear how the proposed methods would handle complex compound concepts.
>
> **A5:**
> Currently, our proposed methods can handle scenarios where multiple concepts contribute to a single class's prediction. For example, our methods handle complex concepts like “a flatbed for carrying cargo” in Figure 7 (p.21) and “a second hand or digital timer” in Figure 8 (p.22). More examples of the concepts are reported in Table 28 (p.32) . IG-CL's average pooling and classification weights, and IN2's prediction layer W_F, are designed to achieve this. We acknowledge that a single unit may represent different concepts, which supports our statement in **A2** that our original IN2 is better. Adding duplicate text concepts that represent different “visual” concepts make IN2 more versatile
>
> > **Q6:** To better clarify the relationship and differences between the proposed methods, the authors should provide more detailed explanations. For instance, adding similar weight visualization for the first method would enable clearer comparisons between concept units and CBM concept sets.
>
> **A6:**
> As we described in the Introduction (p.2) and Section 4 (p.6) in the original submission, IG-CL and IN2 bridge interpretability and continual learning from two perspectives. IG-CL brings interpretability to continual learning by using external interpretability tools to guide models. On the other hand, IN2 tailor interpretable models like CBM for continual learning by leveraging their own interpretability. Both methods aim to integrate interpretability into continual learning, but the interpretability comes from different sources. We add this description to Introduction in the revision. (p.2)
>
> Following your suggestion, we add the visualization of IG-CL’s classification layer in the Figure 6 of Appendix A.1 (p.20) in the revision. For example, for the class “caterpillar”, IG-CL preserves concepts “field” and “dotted”; IN2 in Figure 5 (p.11) preserves ”grass”, “leaves” and “trees”. Compared with IN2, IG-CL can still preserve related concepts, but the concept is more general since IG-CL cannot control the emergence and types of concepts. IN2’s preserved concepts are more task-specific.
>
> **Summary**
>
> To summarize, we have:
> * Described in **A1** for the recommendation of IG-CL’s strategies.
> * Clarified in **A2** for why adding the same concepts when learning new tasks, and doing ablation studies for removing duplicate concepts. Meanwhile, we added some example concepts that related to different classes.
> * Mentioned the more-task scenario experiment results in **A3**.
> * Conducted additional experiments for tasks order in continual learning in **A4**.
> * Described how IN2 handle complex compound concepts in **A5**.
> * Clarified the differences between two proposed methods, and added IG-CL’s prediction weight visualization in **A6**.
>
> We believe we have addressed all your concerns. Please let us know if you still have any reservations and we would be happy to address them.

---

### Decision · Action_Editor_SYPL · 2024-08-12

**Recommendation:** Accept as is

**Comment:**

The reviewers agree that this paper meets the standards of claims and evidence and audience for TMLR, and makes interesting connections between interpretability and continual learning, thus I recommend accepting the paper. There were a few remaining concerns about clarity and density of the paper. In preparing the camera-ready version, I would encourage the authors to further clarify/emphasize some of the details that they added in the revision; for example, promoting some of the methods details from Appendix D.2.2 to the main text—such as how the concepts are generated—would help the reader to understand the work without needing to flip back and forth between the main text and the appendix.

The reviewers also highlighted some lingering concerns, for example about the novelty and especially the limited scope of experiments (i.e., restricted to small image datasets with simple structures, that may lend itself to simple concept-driven methods like these); I concur and therefore I am not recommending the paper for further certification.

**Audience:**

The reviewers agree that the paper will be of interest to some subsets of the TMLR community, by making links between methods for interpretability and continual learning.

**Claims And Evidence:**

The reviewers agree that the revised version of the paper meets the standards of TMLR's claims and evidence, within the scope of the experiments considered.